# Impact of bimetallic interface design on heat generation in plasmonic Au/Pd nanostructures studied by single-particle thermometry

Julian Gargiulo[1,2,3] ✉, Matias Herran [1], Ianina L. Violi [3], Ana Sousa-Castillo[1], Luciana P. Martinez [2], Simone Ezendam[1], Mariano Barella[2,8], Helene Giesler[4], Roland Grzeschik [4], Sebastian Schlücker [4], Stefan A. Maier [1,5,6], Fernando D. Stefani [2,7] & Emiliano Cortés [1] ✉

Localized surface plasmons are lossy and generate heat. However, accurate measurement of the temperature of metallic nanoparticles under illumination remains an open challenge, creating difficulties in the interpretation of results across plasmonic applications. Particularly, there is a quest for understanding the role of temperature in plasmon-assisted catalysis. Bimetallic nanoparticles combining plasmonic with catalytic metals are raising increasing interest in artificial photosynthesis and the production of solar fuels. Here, we perform single-particle thermometry measurements to investigate the link between morphology and light-to-heat conversion of colloidal Au/Pd nanoparticles with two different configurations: core–shell and core-satellite. It is observed that the inclusion of Pd as a shell strongly reduces the photothermal response in comparison to the bare cores, while the inclusion of Pd as satellites keeps photothermal properties almost unaffected. These results contribute to a better understanding of energy conversion processes in plasmon-assisted catalysis.

Plasmonic metals are used in a broad range of applications due to their ability to efficiently confine and manipulate light into a subwavelength scale[1,2]. Most plasmonic materials are intrinsically lossy, meaning that their use is always accompanied by the generation of heat[3,4]. In the so-called field of thermoplasmonics, thermal effects have been exploited in a varied range of applications[5,6]. Despite this, quantitative and accurate experimental measurements of the generated nanoscale thermal fields remain a challenge, hindering the understanding of the mechanisms behind experimental results[7,8].

One area of research where reliable measurements of temperature are of special concern is plasmon-assisted catalysis[9]. Processes associated with plasmon excitation and decay can be beneficial for the chemical transformation of nearby molecules in a variety of ways[10–12]. These include charge carriers generation and injections, enhancement

[1]Nanoinstitute Munich, Faculty of Physics, Ludwig-Maximilians-Universität München, 80539 München, Germany. [2]Centro de Investigaciones en Bionanociencias (CIBION), Consejo Nacional de Investigaciones Científicas y Técnicas (CONICET), C1425FQD Ciudad Autónoma de Buenos Aires, Buenos Aires, Argentina. [3]Instituto de Nanosistemas, Universidad Nacional de San Martín, B1650 Buenos Aires, Argentina. [4]Physical Chemistry I, Department of Chemistry and Center for Nanointegration Duisburg-Essen (CENIDE), University of Duisburg-Essen, 45141 Duisburg-Essen, Germany. [5]School of Physics and Astronomy, Monash University, 3800 Clayton, Australia. [6]Department of Physics, Imperial College London, SW7 2AZ London, UK. [7]Universidad de Buenos Aires, Facultad de Ciencias Exactas y Naturales, Departamento de Física, C1428, Ciudad Autónoma de Buenos Aires, Argentina. [8]Present address: Department of Physics, University of Fribourg, CH-1700 Fribourg, Switzerland. ✉e-mail: jgargiulo@unsam.edu.ar; emiliano.cortes@lmu.de

of the electromagnetic fields, resonant near-field energy transfers, and heating. All these phenomena can occur simultaneously under illumination, complicating the mechanistic understanding of energy flow during plasmon-assisted catalysis. In particular, understanding the role of heat is critical since the rates of chemical reactions usually present an exponential dependence on the temperature, as dictated by Arrhenius law[13]. Therefore, small temperature variations could lead to significant changes in the reaction rates. For this reason, significant efforts are being done to disentangle thermal and non-thermal effects in plasmon-assisted catalysis[14–19], and the problem of reliable assessment of temperature has been identified as one of the bottlenecks for the development of the field[20–22]. A deeper understanding and control of the mechanisms that accelerate reactions would be valuable to achieve superior efficiencies, milder reaction conditions (under solar irradiances and without bulk-scale heating or pressuring of the reactor)[23], or the manipulation of reaction pathways[24,25].

Despite the high potential of plasmon-assisted catalysis, traditional plasmonic metals are intrinsically poor catalysts for many reactions of interest. This has motivated the quest for nanocomposites combining plasmonic metals with conventional catalytic metals (such as Pd, Pt, Rh, etc.)[26–28] The inclusion of a second metal alters the frequency, quality factor, and decay pathways of the plasmon resonances. This allows the tuning of absorption to scattering ratio[28], the spatial distribution of reactivity[29,30], and reaction selectivity[31]. In recent years, several morphological configurations of multimetallic plasmonic nanoparticles (NPs) have been tested, such as spherical core-shell nanoparticles (CS-NPs), nanocubes[32], partially coated NPs[33], alloys[34], core-satellites[35,36], and core-shell trimers[37], demonstrating promising results in artificial photosynthesis and the production of solar fuels[38,39]. However, the addition of other materials to a plasmonic NP also alters the processes of light absorption and heat conduction, in non-trivial ways depending on the overall material composition and its spatial distribution[40–44]. For example, the complexity of heat transport at surface facets and interfaces of hybrid NPs makes their photothermal modeling particularly challenging[45]. Here, this challenge is addressed experimentally, using hyperspectral anti-Stokes (AS) thermometry.

AS thermometry exploits the fact that, upon illumination with a CW laser, the AS part of the photoluminescence (PL) spectrum of plasmonic NPs shows a temperature dependence[46]. This technique has been used for the thermal characterization of individual monometallic Au NPs such as disks[47], pyramids[48], rods[49,50], bowties[51], spheres[52,53], and cylinders[54]. Various implementations of AS thermometry have been reported, as reviewed by Baffou[8]. Recently, our group introduced an implementation called hyperspectral AS thermometry, which retrieves the photothermal coefficient of individual NPs from a single PL hyperspectral image[53]. Unlike other optical methods for single-particle thermometry of metals[3], such as the ones based on fluorescence[55], Raman[56], DNA-PAINT[57,58], optical rotation[59], or refractive index variations[60], hyperspectral AS thermometry is non-invasive, label-free, and does not require any extra characterization or prior knowledge about the NPs or the medium properties.

Here, the optical and photothermal properties of bimetallic plasmonic NPs are studied. First, Au@Pd CS-NPs with different shell thicknesses are designed to investigate changes in their on-resonance photothermal response, understood as the NPs' temperature increase under illumination. We observed that a Pd shell strongly reduces heat generation in comparison with naked Au cores. The experimental studies are complemented with an analytical model for the photothermal response of CS-NPs. Second, the effect of the geometric configuration of bimetallic plasmonic NPs is assessed. Au NPs were combined with Pd in two different configurations: Au@Pd CS-NPs and Au core assembled with Pd satellites. Although the amount of Pd and the plasmon frequency of these two systems are similar, we found that the light absorption efficiency is greater when the Pd is distributed as satellites than when it is confined to the Au core surface as a shell.

These results shed light on the role of the bimetallic interfaces in photothermal heat generation, which is an essential step in achieving a better understanding of photon-phonon conversion processes of plasmonic-catalytic hybrids.

## Results

### Au@Pd core-shell nanoparticles

Figure 1a illustrates the colloidal nanoparticles used in this first part of the study. They consist of Au nanospheres (Au NS) with a diameter of 67 nm, Au@Pd CS-NPs with an Au core of 67 nm, and Pd shells of ≈2 and ≈4 nm in thickness. The NPs were prepared using a seed-mediated growth strategy[35]. This method starts with the synthesis of small Au seed NPs and the subsequent growth to the desired size (see Methods). Two different shell thicknesses were obtained by varying the ratio of Au core and Pd salt. The three colloids are stabilized in water by means of cetyltrimethylammonium chloride (CTAC) capping. Figure 1b shows transmission electron microscopy (TEM) images of the obtained colloidal NPs. Au NSs are spherical, while the Au@Pd are more faceted. The spatial distribution of the two metals measured by energy dispersive X-ray spectroscopy (EDX) is presented as an inset in the bottom panel of Fig. 1b. While the signal of Au is confined to the core region, the Pd signal is uniformly distributed around the entire particle, as expected for a homogeneous coverage. Particle size histograms were obtained from TEM images and are shown in Fig. 1c. The median diameter of the Au NS is (66.6 ± 0.2) nm with a standard deviation of 1.3 nm, while the two CS-NPs colloids have median sizes of (71.5 ± 0.1) nm with a standard deviation of 1.2 nm and (73.8 ± 0.2) nm with a standard deviation of 1.7 nm. These values correspond to median Pd thicknesses of (2.4 ± 0.3) nm, and (3.6 ± 0.3) nm, respectively. In addition, inductive coupled plasma–atomic emission spectroscopy (ICP-AES) was employed to quantify the Pd to Au mass ratio of each CS-NPs colloid. Using this information, the thicknesses of the Pd shells were estimated to be (1.8 ± 0.4) nm and (3.4 ± 0.4) nm (see Supplementary Note 1). For simplicity, we refer to the colloids as Au67 NS, Au67@Pd2, and Au67@Pd4. These parameters were employed to simulate the optical properties of the NP (see Supplementary Note 2). Figure 1d shows the numerically calculated absorption cross-section spectra for different Pd thicknesses, in a homogeneous water environment. The Pd shell leads to significant damping of the plasmon resonance of the Au core, reducing its amplitude and broadening its bandwidth. In addition, a small blue shift of the resonant frequency is predicted. In previous reports, a blue shift is observed for CS-NPs[35], and a redshift for Au nanorods coated with Pd on the tips[61]. Fig. 1e shows the normalized experimental extinction spectra of the three synthesized colloids. The predicted broadening of the resonances due to the Pd shell is clearly observed. However, instead of a blue shift, the spectra exhibit slightly red-shifted resonances, which could also be attributed to the larger concentration of CTAC used as the stabilizing agent of the CS-NPs[62].

To study optical and photothermal properties at the single-particle level, arrays of well-separated individual NPs were fabricated on glass substrates using optical printing, as schematically shown in Fig. 2a[63–65]. Exemplary dark-field images of the optically printed grids are displayed in Fig. 2b. Having ordered arrays of isolated NPs facilitates the automation of data acquisition and allows the collection of larger datasets. It must be noted that the printing process does not alter the stability or composition of the NPs (see Supplementary Note 3). Figure 2c shows representative normalized scattering spectra for each type of NP. CS-NPs present a redshift and a broadening of their resonance with respect to the Au NS cores (see Supplementary Note 4). Because PL emission can provide insights into plasmon decay processes, it is relevant to investigate the emission properties of the NPs under study. Figure 2d shows PL spectra of the types of NPs, excited with CW laser excitation at 532 nm. This wavelength is suited to excite the plasmon resonance of the three kinds of NPs. In agreement with

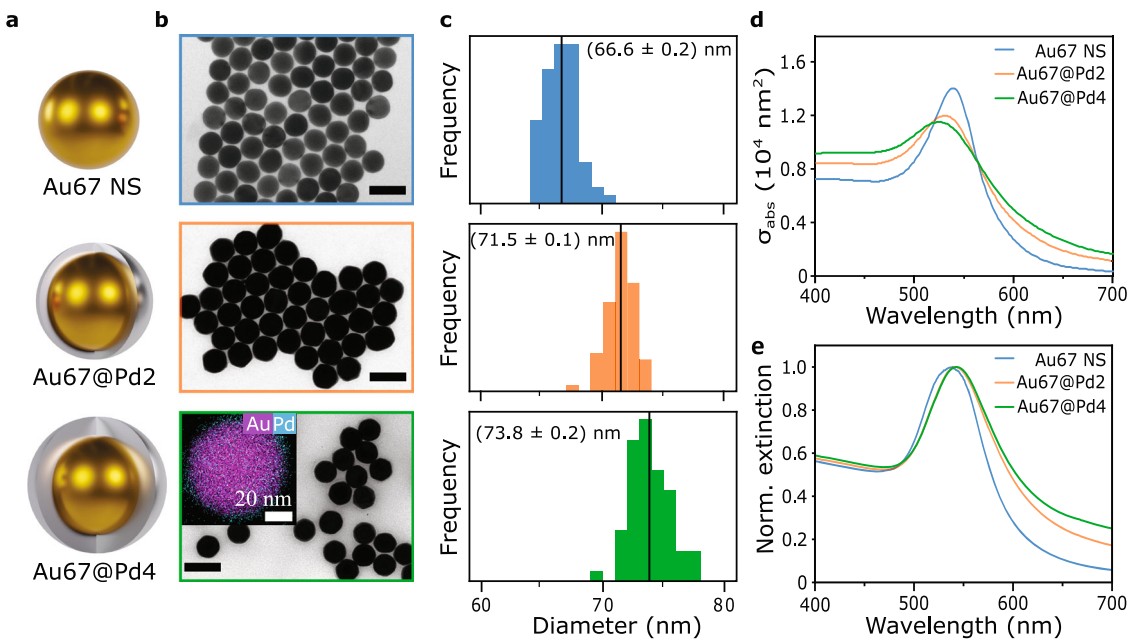

**Fig. 1 | Synthesis and characterization of core-shell Au@Pd nanoparticles.**
**a** Visual illustration and **b** TEM images of the three different studied NPs. Scale bars: 100 nm. The inset shows an EDX with the spatial distribution of the two metals. Au is shown in magenta and Pd in cyan. **c** Size distributions. Medians are indicated with a black line and displayed on the labels. Sample sizes are $N > 60$ in all cases.

**d** Calculated absorption cross-sections in water. **e** Experimental extinction spectra of the colloids. Each spectrum is normalized using its own maximum. Blue, orange, and green correspond to Au67 NS, Au67@Pd2, and Au67@Pd4, respectively. Source data are provided as a Source Data file.

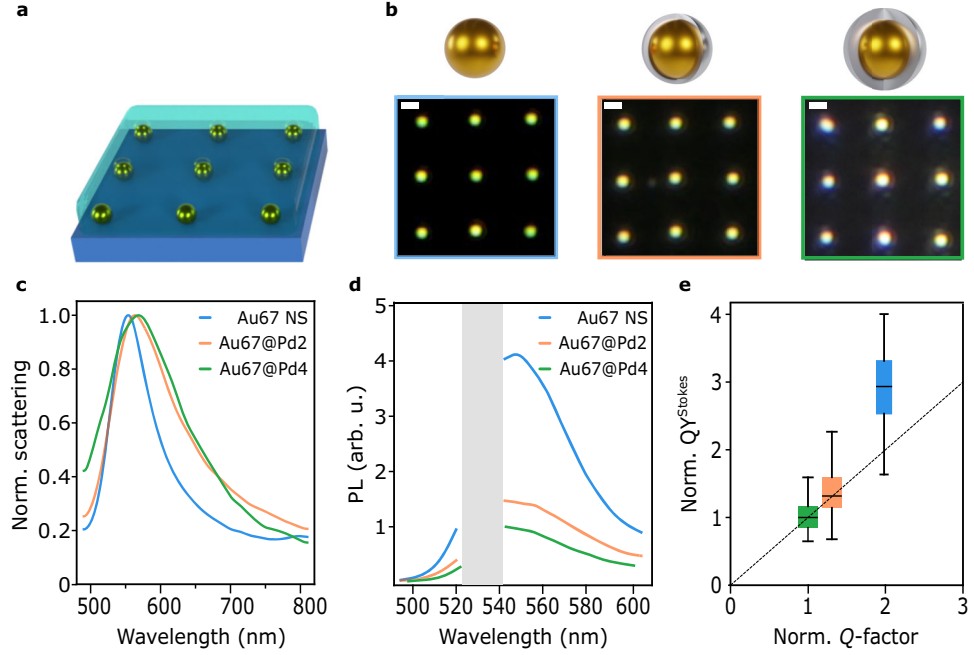

**Fig. 2 | Optical characterization of single Au@Pd core-shell nanoparticles.**
**a** Illustration of fabricated NPs grids on a glass substrate immersed in water. **b** Dark-field images of optically printed grids of individual NPs. Scale bars: 1 μm. **c** Average of single-particle scattering spectra, normalized to their maximum. **d** Average of single-particle PL emission spectra, excited with 1 mW μm⁻² of laser light at 532 nm. A gray band with no data corresponds to the laser rejection filter. **e** Stokes PL

emission quantum yield $QY^{Stokes}$ versus average $Q$-factor. Both magnitudes have been normalized by the measured value for the Au67@Pd4 system. Sample sizes are $N = 390$, $N = 166$, and $N = 122$ for Au67 NS, Au67@Pd2, and Au67@Pd4 respectively. Error bars indicate standard deviations. Blue, orange, and green correspond to Au67 NS, Au67@Pd2, and Au67@Pd4, respectively. Source data are provided as a Source Data file.

reports on other NPs, we found that the shape of the PL spectra follows the characteristics of the scattering ones[66–69]. Interestingly, the amplitude of the PL emission decreases with increasing Pd thicknesses. Qualitatively, this decrease in PL emission is a result of the Pd damping

the plasmon resonance of the Au core[61]. Increased damping could be due to surface effects (increased electron-surface scattering, plasmon decay into interfacial states) or bulk effects (ohmic damping in the Pd shell)[70–72]. For quantitative analysis, the Stokes emission quantum yield

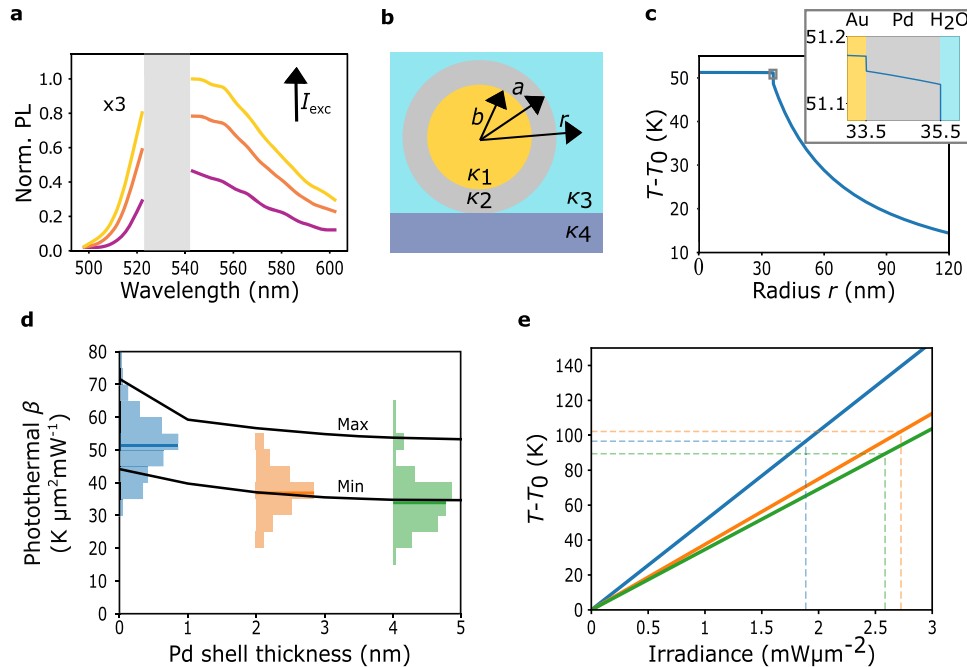

**Fig. 3 | Photothermal properties of single Au@Pd core-shell nanoparticles.**
**a** Exemplary emission PL spectra of a single Au67@Pd2 NP at the following excitation irradiances. $I_{exc} = 1.27$ mW $\mu m^{-2}$ (violet), $I_{exc} = 2.19$ mW $\mu m^{-2}$ (orange) and $I_{exc} = 2.78$ mW $\mu m^{-2}$ (yellow). The AS part has been multiplied by a factor of 3. A gray band with no data corresponds to the laser rejection filter. **b** Illustration of the different constants used in Eq. 2. $r$ is the radial coordinate, $a$ is the radius of the core, $b$ is the external radius of the shell, and $\kappa_1$, $\kappa_2$, $\kappa_3$, and $\kappa_4$ are the thermal conductivities of the **c**ore, shell, medium, and substrate, respectively.
**c** Temperature versus radial coordinate $r$ for an Au67@Pd2 CS-NPs under an irradiance of $I_{exc} = 1$ mW $\mu m^{-2}$. For details on the calculations, see Methods. The inset shows a zoom on the region enclosed with a gray box, corresponding to the interface between the materials. **d** Histograms of the experimentally measured $\beta$ for the three systems under study; blue: Au67 NS, orange: Au67@Pd2, green: Au67@Pd4. The number of measured NPs were $N = 242$, $N = 135$, and $N = 87$ for Au67 NS, Au67@Pd2, and Au67@Pd4 respectively. Solid lines indicate the maximum and minimum theoretical calculation of photothermal coefficient $\beta$ versus thickness of the Pd shell using Eq. 2. Parameters used in the calculus are listed in Methods. **e** Temperature increase versus irradiance for the three studied systems; blue: Au67 NS, orange: Au67@Pd2, green: Au67@Pd4. The dashed lines correspond to the maximum irradiances used in the experiments. In all calculations and experiments here, the NPs are immersed in water and irradiated at 532 nm. Source data are provided as a Source Data file.

$QY^{Stokes}$ was calculated as the ratio between the total Stokes PL emission and the absorbed photons $QY^{Stokes} = \frac{\int_{543}^{\infty} PL(\lambda)d\lambda}{\sigma_{abs}(\lambda = 532nm)I_{exc}}$, where $I_{exc}$ is the excitation irradiance (see Supplementary Note 5). The value of $\sigma_{abs}(\lambda = 532$ nm) was simulated for the different NPs on glass substrates and water environments (see Supplementary Note 2). In addition, the resonance quality was quantified as $Q$-factor $= \frac{E_{res}}{\Gamma}$ where $E_{res}$ is the resonance energy and $\Gamma$ its full width at half maximum calculated from the measured scattering spectra. Figure 2e shows the measured $QY^{Stokes}$ versus $Q$-factor for the three NPs, showing a positive correlation. A similar trend was reported for Au nanorods and ascribed to enhanced emission due to an increased density of photonic states (Purcell effect)[61,73–75].

After the optical characterization of the individual NPs, their photothermal response was measured using hyperspectral anti-Stokes thermometry with a CW 532 nm laser. The technique, as introduced by ref. 53, allows single-particle photothermal characterization by scanning a laser over the NP (see Supplementary Fig. 9a). In this way, heating and PL excitation are performed simultaneously with the same beam. Throughout the scanning, the relative position between the beam and the NP changes, leading to different excitation irradiances and hence, different steady-state temperatures. For each NP, a square raster scanning consisting of $10 \times 10$ pixels on a range of $0.8 \times 0.8$ $\mu m^2$ is performed. On each pixel, the PL spectra is measured with an integration time of 2.5 s. In such a manner, a set of one hundred temperature-dependent PL emission spectra is collected. Figure 3a shows three PL spectra for different irradiance levels for an illuminated Au67@Pd2 CS-NP. Processing the set of acquired PL spectra allows for finding the photothermal coefficient $\beta$, defined as

$$T^{NP} = \beta I_{exc} + T_0 \tag{1}$$

where $T^{NP}$ and $T_0$ are the temperature of the particle in the presence and absence of light, respectively. For details on data acquisition and processing, see Supplementary Note 6.

It must be noted that this method assumes that the PL-emitting object has a homogeneous temperature $T^{NP}$. This condition is fulfilled by Au NS under CW illumination[3], and is also true for Au@Pd CS-NPs, as we demonstrate in the following. Figure 3c shows the calculated temperature increase $T(r) \cdot T_0$ versus the radial coordinate $r$ for an Au67@Pd2 NP immersed in water under illumination at 532 nm with an irradiance of $I_{exc} = 1$ mW $\mu m^{-2}$ (see Fig. 3b for the definition of the parameters and Supplementary Note 7 for a full derivation of the calculus). The temperature is practically constant inside the CS-NP, except for a small drop $\Delta T_{Au-Pd}$ at the Au/Pd interface, shown in the inset of Fig. 3c. The temperature variations inside the CS-NP are $10^{-3}$ times smaller than the temperature increase of the NP surface. Hence, the NP can be described by a single, uniform temperature $T^{NP}$. This is a consequence of the high thermal conductivities of Au and Pd with respect to water, and the high electronic thermal conductance of the Au/Pd interface that allows efficient heat transfer through electron-electron scattering[76]. In addition, because Au and Pd have similarly high volumetric electron-phonon couplings, electrons and phonons equilibrate rapidly within each metal. For this reason, $T^{NP}$ refers indistinctly to the electronic or lattice temperature. See Supplementary Note 8 for a quantitative comparison between lattice and electron gas temperatures. Thus, AS thermometry retrieves the photothermal

coefficient $\beta$ of the entire CS-NPs. In this work, $\beta$ will be used as a metric to quantify photothermal responses at the single-particle level. For a discussion on different metrics commonly used in the literature see Methods.

Figure 3d shows histograms of the measured photothermal coefficients for the three systems under study. The median value obtained for the Au NS is $\beta_{Au67}$ is $(51 \pm 1)$ K μm² mW⁻¹, in line with the one reported by Barella et al. for 64 nm Au NSs[53]. Interestingly, a significant reduction of $\beta$ is observed for the CS-NPs. The obtained median values of the Au67@Pd2 and Au67@Pd4 were $\beta_{Au67@Pd2} = (38 \pm 1)$ K μm² mW⁻¹ and $\beta_{Au67@Pd4} = (35 \pm 1)$ K μm² mW⁻¹, respectively. The standard deviation of the three measurements is $\sigma = 8$ K μm² mW⁻¹. Figure 3e (solid lines) shows the temperature increase versus excitation irradiance following Eq. 1, corresponding to the median $\beta$ of each type of NP. In dashed lines, the maximum irradiances used in each experiment are shown. The maximum temperature reached by the NPs was around 100 °C. It must be noted that at these temperatures, the NPs are stable and no changes in their scattering or PL spectrum were observed during or after the measurements (see Supplementary Note 3).

An analytical model was developed to calculate the temperature $T^{NP}$ of a spherical NP of radius $a$ surrounded by a media of thermal conductivity $\kappa_3$ and supported on a substrate with thermal conductivity $\kappa_4$. For details, see Methods and Supplementary Notes 7 and 8. The temperature $T^{NP}$ is given by

$$T^{NP} = f\left(\frac{1}{4\pi\kappa_3 a} + \frac{R_{2-3}^{th}}{4\pi a^2}\right)\sigma_{abs}I_{exc} + T_0 \qquad (2)$$

where $R_{2-3}^{th}$ is the Kapitza interfacial thermal resistances between the shell and the surrounding media, and $f = \left(1 - \frac{\kappa_4 - \kappa_3}{2(\kappa_4 + \kappa_3)}\right)$ is a factor that accounts for the role of the substrate in the heat dissipation (see Supplementary Note 9). The solid black lines in Fig. 3d correspond to the photothermal coefficient $\beta$ predicted by Eq. 2 for Au@Pd CS-NPs as a function of the Pd shell thickness, for NPs on glass, surrounded by water and illuminated at 532 nm, as in the experiments. The calculation requires several thermodynamical constants for which accurate experimental values are scarce. This is the case for example for the Kapitza resistance between Pd and water $R_{Pd-water}^{th}$. For this reason, we have included a maximum and a minimum calculated value, to represent the large dispersions of available data in the literature. A list of used parameters is shown in Table 2. The calculations reasonably predict the experimental trends. However, it must be noted that for Au@Pd, half of the measured CS-NPs had a value below the predicted range. This could be due to several factors. (i) An overestimation of the absorption cross-sections: The simulations (see Fig. 1d and Supplementary Fig. 3) predict a 10 nm blue shift in the resonant frequency of CS-NPs, which is not observed experimentally. However, considering that the resonances are broad, a 10 nm detuning of the spectrum versus the excitation wavelength only modifies the absorption cross-sections by less than 2% (ii) The influence of the surfactant CTAC in the thermal resistance of the Pd-water interface[77]. (iii) an overestimation of the factor $f$ accounting for the effect of the substrate in heat dissipation. For spherical NPs immersed in water on a glass substrate, $f$ takes a value of 0.875. However, Au@Pd CS-NPs are faceted, as shown in Fig. 1b, and can present a larger contact area with the substrate, enhancing heat dissipation. Thermal simulations estimate a value of $f = 0.843$ for faceted NPs, which is 4% smaller than its spherical counterpart (see Supplementary Note 10). To summarize, accurate theoretical predictions require a complete knowledge and modeling of every geometrical boundary, also including the liquid-substrate interface. Having a precise description of all these factors is challenging, making most predictions only approximate, and

reinforcing the need for methods able to measure the temperature of nanoscale objects in their operation environments (i.e., in situ).

## Impact of the bimetallic interface on heat generation

The functionalities of bimetallic nanostructures are not only determined by the material composition, but also by the spatial distribution of the constituents[35]. In the following, the effect of morphology on light-to-heat conversion for different Au-Pd bimetallic structures is investigated. Au NS 60 nm diameter cores were combined with Pd in two different configurations: (i) assembled with spherical Pd satellites NPs (named Au60-Pd-sat). The satellites have a median diameter of $(6.2 \pm 0.3)$ nm and a standard deviation of 1.3 nm. (ii) coated with a homogeneous $(1.8 \pm 0.3)$ nm Pd shell (named Au60@Pd2). Figure 4a shows an illustration of the three synthetized systems, as well as their corresponding TEM images. Detailed characterizations of the three systems are provided in Supplementary Note 1 and Supplementary Note 11. The average number of satellites per NP of the Au60-Pd-sat was estimated to be $\langle N_s \rangle = (54 \pm 7)$. ICP-AES was employed to quantify the Pd to Au mass ratio $\frac{M_{Pd}}{M_{Au}}$ of each system, yielding the values $\frac{M_{Pd}}{M_{Au}} = (0.073 \pm 0.004)$ and $\frac{M_{Pd}}{M_{Au}} = (0.036 \pm 0.002)$, for Au60@Pd2 and Au60-Pd-sat respectively. This means that the amount of Pd is on the same order of magnitude for both configurations, being the total mass of Pd in the satellites being approximately half the mass in a 2 nm Pd shell. Figure 4b shows the normalized extinction spectra of the three studied systems. Again, adding Pd to the Au cores results in plasmon damping and a decrease in the resonance quality, although significantly larger damping was observed for the CS-NP. While the $Q$-factor of the Au60@Pd2 system is reduced by ≈27% in comparison with the Au cores, it is only reduced by ≈3% for the Au60-Pd-sat. This can be explained by the significantly lower contact area between Au and Pd in the Au60-Pd-sat system in comparison with the Au60@Pd2 (see Supplementary Note 12).

Arrays of individual Au NS 60 nm and Au60@Pd2 CS-NPs were fabricated through optical printing on glass substrates. Instead, the Au60-Pd-sat were deposited by a drop-casting method, because the solution was not stable enough for the optical printing process. Figure 4c shows representative single-particle PL emission spectra when excited with laser light at 532 nm in a water environment. The lower $Q$-factor is reflected in a drop in the single-particle PL emission. The decrease in PL emission is significantly larger in the Au60@Pd2 CS-NPs compared to the Au60-Pd-sat, in concordance with a larger plasmon damping. It must be noted that the Pd satellites interact weakly with light (see Supplementary Fig. 19) and absorb only a minor fraction of the incoming light (altogether less than 10%, when placed around the core)[35]. Hence, the PL emission of the Au60-Pd-sat system is mostly emitted by the Au core.

Next, the photothermal coefficient of each system was determined using hyperspectral AS thermometry. The resulting histograms are presented in Fig. 4d. The median photothermal coefficient for the Au 60 NS cores was $\beta_{Au60} = (47 \pm 1)$ K μm² mW⁻¹ with a standard deviation of 8 K μm² mW⁻¹. There is no significant difference with the median value for the core-satellites $\beta_{Au60-Pd-sat} = (46 \pm 1)$ K μm² mW⁻¹, with a larger standard deviation of 13 K μm² mW⁻¹. If the PL is mostly emitted by the Au, the measured $\beta$ corresponds to the temperature of the core. This result can be explained by a small reduction of the absorption cross-section of the Au60-Pd-sat system with respect to the Au cores (see Supplementary Note 12). By contrast, a significant reduction of the photothermal coefficient is observed for the CS-NPs, with a median of $\beta_{Au60@Pd2} = (29 \pm 1)$ K μm² mW⁻¹ and a standard deviation of 5 K μm² mW⁻¹, in line with the results presented in the previous section of this work. Overall, the experiments of this section point out that the presence of an interface between the two metals is crucial as it dictates the photothermal response of the bimetallic nanostructures. The direct contact between Au and Pd enhances the

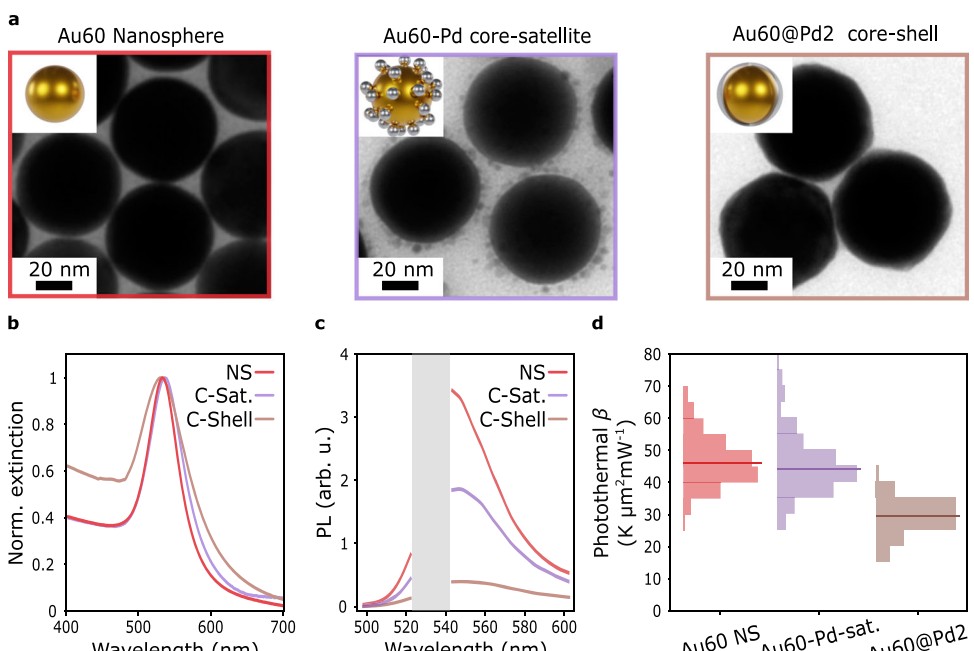

**Fig. 4 | Optical and photothermal comparison between core-shell and core-satellites AuPd nanoparticles. a** Illustration and TEM images of the synthesized NPs. **b** Experimental extinction spectra. Each spectrum is normalized using its own maximum. **c** Average of single-particle PL emission spectra of NPs on glass substrates and immersed in water, excited with 1 mW μm⁻² of laser light at 532 nm. A gray band with no data corresponds to the laser rejection filter. **d** Histograms of the experimental measured photothermal coefficient $\beta$ for the three systems on glass substrates and immersed in water, at 532 nm. Red, violet, and brown correspond to Au60 NS, Au60-Pd-sat, and Au60@Pd2, respectively. Source data are provided as a Source Data file.

**Table 1 | Summary of optical and photothermal properties of Au/Pd NPs**

|  | Diameter (nm) | Extinction $\lambda_{max}$ (nm) | Scattering $\lambda_{max}$ (nm) | Scattering FWHM (nm) | $\frac{QY^{Stokes}}{QY^{Stokes}_{Au67@Pd4}}$ | Photothermal $\beta$ (K μm² mW⁻¹) |
|---|---|---|---|---|---|---|
| Au67 NS | 66.6 ± 0.2 | 535 | 551 ± 3 | 80 ± 3 | 2.8 | 51 ± 1 |
| Au67@Pd2 | 71.4 ± 0.3 | 543 | 561 ± 2 | 119 ± 6 | 1.3 | 38 ± 1 |
| Au67@Pd4 | 73.8 ± 0.3 | 543 | 565 ± 3 | 155 ± 16 | 1 | 35 ± 1 |
| Au60 NS | 59.7 ± 0.2 | 532 | 547 ± 2 | 77 ± 3 | 2.3 | 47 ± 1 |
| Au60-Pd-sat | 59.7 ± 0.2[a] | 535 | 547 ± 2 | 79 ± 3 | 1.4[b] | 46 ± 1 |
| Au60@Pd2 | 63.3 ± 0.3 | 530 | 551 ± 2 | 106 ± 9 | 0.7 | 29 ± 1 |

The photothermal coefficient $\beta$ is indicated at the measured wavelength of 532 nm. The diameters were measured from TEM images.
[a]For the Au60-Pd-sat system, the diameter in the table corresponds to the core size, without considering the satellites.
[b]In the calculations of the absorption cross-section, a core-satellite gap size of 1 nm was used, as described in Supplementary Note 12.

surface damping of the plasmon, leading to a poorer quality factor and lower absorption.

## Discussion

The optical and photothermal properties of individual bimetallic gold-palladium NPs in core-shell and core-satellite configurations were measured on glass substrates surrounded by water. Measured properties are summarized in Table 1. Using AS thermometry, it was possible to determine the temperature increases at the surface of the Au-Pd NPs, in opposition to previous studies focused on the bulk (i.e., macroscopic) heating of solutions (see Methods section for a discussion on this topic). Photothermal coefficients were measured at an excitation wavelength of 532 nm, close to the NPs plasmon resonances of all systems. First, Au@Pd CS-NPs of 67 nm Au core and two different Pd thicknesses were studied. The inclusion of a Pd shell leads to poorer quality factors as evidenced by the broadening of scattering spectra and lower Stokes PL emission quantum yield. This is attributed to the enhanced surface damping of the plasmon in the Au-Pd interface and causes a decrease in the absorption cross-section. Using AS thermometry, we found that the inclusion of a 2-nm thick Pd shell leads to a reduction of the photothermal coefficient $\beta$ of 26% with respect to the bare Au core. However, the thickness of the Pd shell does not have a strong influence. Only a 6% decrease in the photothermal coefficient was observed when the Pd thickness was increased from 2.4 to 3.6 nm. Since Pd is highly conductive, it does not significantly restrict the conduction of heat to the surroundings, meaning that the absorption cross-section of the Au-Pd CS-NPs is the main parameter influencing its photothermal coefficient. This conclusion can be extended to other CS-NPs where both materials are highly conductive. (See Supplementary Note 14 for a discussion on Au@Pt, Au@Rh, and Au@Ag NPs). In addition, an analytical model was introduced for monometallic NS and bimetallic CS-NPs. Remarkably, this simple model predicts reasonably well the magnitude of the measured photothermal coefficients and their experimental trend and is applicable to other bimetallic CS-NPs systems. However, it slightly overestimates the value of $\beta$ for CS-NPs. A more accurate description requires precise information on interfacial thermal resistances, nanoscale boundaries, and the consideration of the liquid-substrate interaction. This level of description is challenging to achieve, highlighting the benefits of the experimental approach presented here.

Secondly, the influence of the Au-Pd interface on the photothermal properties was studied. It was found that coating the Au 60 nm

cores with a 1.8 nm homogeneous shell of Pd reduces the photo-thermal coefficient by 38%, but that it remains practically unaffected when a similar amount of Pd is included in small satellites.

Overall, it can be concluded that the inclusion of Pd affects the photothermal coefficient of Au NPs mainly by altering their absorption cross-section due to plasmon damping. The main characteristic governing the damping is the presence of an interface between materials, with a minor influence on the total amount of added Pd. In this sense, the experiments presented highlight the importance of interface management for light-to-heat conversion.

Photothermal characterization at the single-particle level enables a better understanding of the link between the NPs structure and the generated heat. As shown here, two systems with the same materials (in composition and quantity) can have very different photothermal behaviors according to the spatial distributions of their constituents. In many cases, higher temperatures are desirable and beneficial in catalysis, despite other non-thermal mechanisms operating simultaneously. The results presented here provide guidelines for better exploitation of thermal fields. For example, they allow a more efficient deposition of the (typically scarce and expensive) catalytic materials on plasmonic systems, for the maximization of the photothermal response. In addition, photothermal characterization can be combined with other studies of the role of morphology and composition in the reactivity of nanosystems[26,78–80] to provide insightful information on energy transfer mechanisms and reaction pathways. For example, in prior studies by Herran et al., it was demonstrated that core-satellite systems show a higher enhancement for hydrogen generation upon illumination, compared to core-shell ones[35]. In this context, AS thermometry provides label-free, non-invasive, and in situ information at the single-particle level, and therefore can potentially be employed under reaction conditions in the liquid or gas phase (as shown here by the measurements done in aqueous media). For this reason, it is envisioned that AS thermometry will be applied to many other complex/hybrid interfaces with catalytic relevance, such as metal-semiconductors or metal-organic materials[39]. More broadly, we believe that nanothermometry can contribute to a better understanding of the role of temperature, not only in catalysis but also in the wide range of applications of hybrid or complex plasmonic materials.

## Methods

### Materials and reagents
Gold (III) chloride trihydrate (HAuCl$_4$·3H$_2$O, ≥99.9%), ascorbic acid (AA, ≥99.0%), sodium borohydride (NaBH$_4$, ≥98%), cetyl-trimethylammonium bromide (CTAB, ≥99.0%), CTAC (25 wt% in water), tetraethylorthosilicate 98% (TEOS), polyvinylpyrrolidone (PVP, $M_W$: 10 kg mol$^{-1}$), ammonium hydroxide solution (NH$_4$OH, 27 wt% in water), and absolute ethanol (EtOH, 99.8%) were all purchased from Sigma-Aldrich and used without further purification. In all experiments, ultrapure water with a resistivity of 18.2 MΩ cm was used.

### Synthesis of 60 and 67 nm Au NS
Au nanoparticles were prepared using a seed-mediated growth strategy described by ref. [81] with some modifications to scale up the reaction. Step 1—Au seed synthesis: The initial Au seeds were formed by reducing 5 mL of an aqueous solution containing 0.25 mM HAuCl$_4$ and 0.1 M CTAB with 300 μL of a freshly prepared 10 mM NaBH$_4$ solution (ice bath). The solution turned brownish within seconds indicating seeds formation. The seeds were kept undisturbed for 3 h at 27 °C. Step 2—10 nm Au NS synthesis: In a 20 mL glass vial, 4 mL of a 200 mM CTAC solution, and 3 mL of a 100 mM AA solution were mixed, followed by the addition of 100 μL of previously prepared CTAB capped Au seeds. After the solution was stirred at 300 rpm and left at 27 °C for 10 min, 4 mL of 0.5 mM HAuCl$_4$ were added in a quick one-shot injection. The resulting 10 nm Au spheres were centrifuged twice

at 12045×$g$ for 30 min and finally redispersed in 1 mL of a 20 mM CTAC solution. Step 3–60 and 67 nm Au NS synthesis: 200 mL of a 100 mM CTAC solution were mixed with 1.3 mL of a 100 mM AA solution and 225 or 250 μL of 10 nm Au NS solution to obtain 67 and 60 nm Au NS, respectively. The whole solution was kept stirred at 800 rpm using stirring bars at 27 °C for 20 min. Then, 10 mL of 10 mM HAuCl$_4$ were pumped in at a rate of 20 mL h$^{-1}$ and once the pumping was completed, was left stirring for 30 min. After two centrifugation-redispersion cycles (2147×$g$, 10 min) the Au NS were finally redispersed in 35 mL of H$_2$O.

### Synthesis of Au@Pd CS-NPs
To grow a 2 nm Pd shell on top of the 60 and 67 nm Au cores, a solution containing 1.785 mL of a CTAC solution, 2.30 mL of a 0.2 M CTAB solution, 400 μL of a 10 mM K$_2$PdCl$_4$ and 13.96 mL H$_2$O were kept at 100 °C for 60 min. Then, 1.753 mL of an aqueous solution containing ≈7.5 × 10$^{11}$ Au NS were added at 100 °C, followed by the quick injection of 1 mL of 100 mM AA solution. The solutions immediately turned purple and were left at 100 °C for another 60 min. Finally, two centrifugation-redispersion cycles (1207×$g$, for 10 min) were carried out. Alternatively, to grow a 4 nm Pd shell on 67 nm Au NS, the same procedure was followed, but using 1.753 mL of an aqueous solution containing 6.0 × 10$^{11}$ Au NS.

The Au67@Pd2 and Au67@Pd4 were redispersed in 20 mM CTAC, while Au60@Pd2 was redispersed in H$_2$O. For more information, see ref. [35].

### Synthesis of Au60−Pd-sat NPs
Step 1—synthesis of Pd satellites: PVP-stabilized Pd NPs were synthesized according to the following method, as previously described by ref. [82]. Briefly, 45 mL of aqueous solution containing 1.05 mM PVP and 4.25 mM of AA was heated up to 100 °C under reflux for 10 min. Subsequently, 5.0 mL containing 10 mM of Na$_2$PdCl$_4$ was added in one shot. The reaction was allowed to continue at 100 °C for 3 h to obtain Pd NPs. Step 2—assembly of Au60-Pd-sat NPs: 4 mL of EtOH were added to 2.77 mL of a 2.72 × 10$^{14}$ Au NS L$^{-1}$ aqueous solution, followed by the addition of 350 μL of non-washed Pd satellites. After stirring the solution at 400 rpm for 30 min, it was washed in low-binding DNA Eppendorf tubes twice at 604×$g$ for 20 min. For more information, see ref. [35].

### Inductively coupled plasma−atomic emission spectroscopy
ICP-AES (Agilent 5800) was used to quantify the material composition of the bimetallic NPs. The sample preparation consisted of dissolving 300 μL of each colloidal solution with 500 μL of fresh reversed aqua regia (HNO$_3$: HCl 3:1), followed by a dilution to 4 mL using HCl 2% wt. The components of the sample, Au and Pd, were identified based on their characteristic emissions ($\lambda_{Au}$ = 242 nm, $\lambda_{Pd}$ = 340 nm). First, a calibration curve between concentration and emission intensity was obtained by preparing 5, 10, 30, and 50 ppm (mg L$^{-1}$) Au and Pd solutions from a 1000 ppm original solution purchased from Sigma-Aldrich. The calibration is shown in Supplementary Fig. 24. Then, the amount of material present in the investigated Au and AuPd NPs was determined from the intensity of emitted light.

### Optical printing
NPs were optically printed onto glass substrates according to the following process: The NP suspension is placed on top of a substrate functionalized with poly-(diallyl-dimethylammonium chloride) (PDDA, Sigma-Aldrich, $M_W$ ≈ 400–500 kg mol$^{-1}$). The process is carried out in an open chamber, employing a 532 nm laser (Ventus, Laser Quantum) and a 60x water-immersion (Olympus) objective with a NA = 1. Printing irradiance was adjusted for each system to optimize the printing time and avoid morphological changes during the optical printing process. Typically, 8 mW μm$^{-2}$ was used to print 60 and 67 nm Au NP. For the

Au67@Pd CS-NPs, the irradiance was set to 6 mW μm$^{-2}$ and for the Au60@Pd it was 5.4 mW μm$^{-2}$. For further details on the fundamentals of optical printing, see refs. [83],[84].

## Dark-field microscopy and spectroscopy

Dark-field images and single NP spectra were acquired by employing a high-intensity visible-NIR light source (Thorlabs, OSL2IR) which was scattered by the sample and focused on 60x water-immersion (Olympus) through a dark field condenser. A digital camera (Canon EOS-500D) was used to obtain dark field images of the NPs supported on the glass substrate. Unpolarized scattering spectra of each NP were measured with a Shamrock 500i spectrometer (Andor) and an EMCCD camera (Andor Ixon EM + 885), in a spectral range between 450 and 850 nm (1 nm resolution).

## Hyperspectral confocal PL images

For confocal hyperspectral PL imaging, a continuous wave 532 nm laser was focused near its diffraction limit with a mean beam waist of 311 nm. Scanning was achieved with a closed-loop piezoelectric stage (PI). Each confocal image was obtained with 10 × 10 pixels on a range of $0.8 \times 0.8\ \mu m^2$, with a high electron-multiplying gain (usually 150) and an integration time of 2.5 s for each pixel. The PL spectra were measured from the 500 to 600 nm range, employing two 532 nm notch filters to filter the laser line. Hyperspectral confocal images of 67 nm NS, Au67@Pd2 CS-NPs, and Au67@Pd4 CS-NPs were acquired using a maximum irradiance of 2.1, 2.8, and 2.7 mW μm$^{-2}$ respectively. For 60 nm Au NS, 60 nm core-satellites AuNP@Pd and Au60@Pd2 CS-NPs, the irradiances were 2.0, 2.4, and 4.9 mW μm$^{-2}$, respectively.

## AS hyperspectral thermometry

The photothermal coefficient was extracted from PL spectra as follows: First, a set of PL spectra corresponding to different excitation irradiances $I_i^{exc}$ is extracted from the hyperspectral images. Then, all possible ratios $Q_{i,j}^{AS}$ between AS PL emissions are calculated and fitted with Eq. 3.

$$Q_{i,j}^{AS}(\lambda) = \frac{I_i^{exc}}{I_j^{exc}} \frac{e^{\frac{E(\lambda)-E(\lambda_{exc})}{k_B\left[T_0+\beta_{i,j}I_j^{exc}\right]}} - 1}{e^{\frac{E(\lambda)-E(\lambda_{exc})}{k_B\left[T_0+\beta_{i,j}I_i^{exc}\right]}} - 1} \tag{3}$$

with the photothermal coefficient $\beta_{i,j}$ the only unknown parameter, $E(\lambda)$ the energy of the photon and $k_B$ the Boltzmann constant. Then, all obtained photothermal coefficients $\beta_{i,j}$ are averaged to obtain a single $\beta$ for the scanned NP. For examples of data analysis and fitting please see Supplementary Note 6. For more information on the fundamentals of the technique see ref. [53].

## Temperature modeling of CS-NPs

The temperature $T$ versus de radial coordinate r of a core-shell NPs is

$$
\begin{aligned}
&r < b \\
&T(r) = \frac{-q_1}{6\kappa_1}r^2 + c_1 \\
&b < r < a \\
&T(r) = \frac{-q_2}{6\kappa_2}r^2 + \frac{c_2}{r} + c_3 \\
&r > a \\
&T(r) = \frac{Q}{4\pi\kappa_3 r} + T_0
\end{aligned}
\tag{4}
$$

A visual description of the parameters is presented in Fig. 3c. A full derivation of Eq. 6 is provided in Supplementary Note 7. $\kappa_1$, $q_1$, and $b$ are the thermal conductivity, heat power density, and radius of the core,

while $\kappa_2$, $q_2$, and $a$ corresponds to the outer radius of the core-shell NP. $T_0$ is the room temperature and $c_1$, $c_2$, and $c_3$ are integration constants given by $c_2 = \frac{\kappa_3}{\kappa_2}\frac{Q}{4\pi\kappa_3} - \frac{q_2 a^3}{3\kappa_2}$, $c_3 = \frac{\kappa_3 R_{2-3}^{th} c_4}{a^2} + \frac{c_4}{a} + \frac{q_2 a^2}{6\kappa_2} - \frac{c_2}{a} + T_0$, and $c_1 = \frac{1}{3}q_1 b R_{1-2}^{th} + c_3 + \frac{c_2}{b} + \frac{q_1 b^2}{6\kappa_1} - \frac{q_2 b^2}{6\kappa_2}$. $R_{1-2}^{th}$, and $R_{2-3}^{th}$ are the Kapitza interfacial thermal resistances between materials and $Q = \sigma_{abs}(\lambda)I_{exc}$ is the total absorbed heat by the NP.

The temperature drop inside the Au volume is

$$\triangle T_{Au} = T(b) - T(0) = \frac{-q_1}{6\kappa_1}b^2 \tag{5}$$

The temperature drop at the Au/Pd interface is

$$\triangle T_{Au-Pd} = T(b^-) - T(b^+) = \frac{1}{3}q_1 b R_{Au-Pd}^{th} \tag{6}$$

The temperature drop in Pd is

$$\nabla T_{Pd} = T(a) - T(b) = \frac{-q_2}{6\kappa_1}(b^2 - a^2) + c_2(\frac{1}{b} - \frac{1}{a}) \tag{7}$$

From these expressions, it can be shown that $\frac{T(a^-)-T(0)}{T(a^+)-T_0} \leq 10^{-3}$, meaning that temperature variations inside the NP are negligible. In that case, the temperature inside the NP becomes

$$T^{NP} = T(a^+) + \frac{Q}{4\pi a^2}R_{2-3}^{th} = Q\left(\frac{1}{4\pi\kappa_3 a} + \frac{R_{2-3}^{th}}{4\pi a^2}\right) + T_0 \tag{8}$$

The presence of a substrate with thermal conductivity $\kappa_4$ will affect heat dissipation around the NP. This effect can be modeled by multiplying Eq. 7 by a heat dissipation factor $f = (1 - \frac{\kappa_4 - \kappa_3}{2(\kappa_4 + \kappa_3)})$ calculated by the image method (see Supplementary Note 8).

The calculation shown in Fig. 3d uses Eq. 6 and the following parameters: $\kappa_1 = 318\ W\ m^{-1}\ K^{-1}$, $\kappa_2 = 71\ W\ m^{-1}\ K^{-1}$, $\kappa_3 = 0.6\ W\ m^{-1}\ K^{-1}$, $R_{2-3}^{th} = 3 \times 10^{-9}\ m^2 K\ W^{-1}$, $R_{1-2}^{th} = 0.03 \times 10^{-9}\ m^2\ K\ W^{-1}$, $\sigma_{abs} = 1.49 \times 10^{-14}\ m^2$, $b = 33.5\ nm$, $a = 35.5\ nm$, $I_{exc} = 1\ mW\ \mu m^{-2}$, $Q = \sigma_{abs}I_{exc}$, $q_1 = 0.7\frac{Q}{\frac{4}{3}\pi b^3}$, $q_2 = 0.3\frac{Q}{\frac{4}{3}\pi(a^3-b^3)}$. The calculation shown in Fig. 3f uses Eq. 2 with the following parameters: $\kappa_1 = 318\ W\ m^{-1}\ K^{-1}$, $\kappa_2 = 71\ W\ m^{-1}\ K^{-1}$, and the values from Table 2 for the maximum and minimum curves.

## Discussion on heating efficiency metrics

In this work, photothermal coefficients $\beta$ are quantified, that allows the calculation of the NP superficial temperature increase $\Delta T_{NP} = T^{NP} - T_0$ by using Eq. 1. However, in many applications, colloidal suspensions are used to heat solutions. In that context, heating is a collective effect of many NPs suspended in a solvent. The relevant parameter to describe such a system is the bulk temperature increase of the solution $\Delta T_s$. In the steady state, it can be calculated as (see ref. [44])

$$\triangle T_s = \frac{Q_{in}}{m_s C_s B} \tag{9}$$

where $m_s$ and $C_s$ are the mass and the heat capacity of the solvent, $Q_{in}$ is the total heat dissipated from the NPs into the solution, and $B$ is the rate of heat dissipation from the solution to the external environment. It must be noted that the bulk temperature increase of the solution $\Delta T_s$ is different (usually, much lower) than the surface temperature increase of the NPs suspended $\Delta T_{NP}$ in the solution.

Typically, $Q_{in}$ is calculated as

$$Q_{in} = (P_0 - P_T)\eta = P_0(1 - 10^{-OD})\eta \tag{10}$$

**Table 2 | Thermodynamic constants used**

| Parameter | Min Value | Reference | Max Value | Reference |
|---|---|---|---|---|
| $R_{\text{Au-water}}^{\text{th}}$ | $4 \times 10^{-9}\,\text{m}^2\,\text{K}\,\text{W}^{-1}$ | 77,85 | $15 \times 10^{-9}\,\text{m}^2\,\text{K}\,\text{W}^{-1}$ | 86 |
| $R_{\text{Pd-water}}^{\text{th}}$ | $1.8 \times 10^{-9}\,\text{m}^2\,\text{K}\,\text{W}^{-1}$ | 87 | $2.4 \times 10^{-9}\,\text{m}^2\,\text{K}\,\text{W}^{-1}$ | 88 |
| $\kappa_3$ | $0.6\,\text{W}\,\text{m}^{-1}\,\text{K}^{-1}$ | $T = 20\,°C^3$ | $0.68\,\text{W}\,\text{m}^{-1}\,\text{K}^{-1}$ | $T = 80\,°C^3$ |
| $\kappa_4$ | $1\,\text{W}\,\text{m}^{-1}\,\text{K}^{-1}$ | | $1.2\,\text{W}\,\text{m}^{-1}\,\text{K}^{-1}$ | 3 |
| $\sigma_{\text{abs}}(\lambda = 532\,\text{nm})$ | see Suppl. Table 2 | | | |
| $a$ | $b$ + Pd thickness | | | |
| $b$ | 63.6 nm | Fig. 1b | 71 nm | Fig. 1b |

where $P_0$ and $P_T$ are the incidents and transmitted powers, $\eta$ is the photothermal efficiency of the solution, and OD is the optical density of the solution. This parameter is defined as the ratio between the total absorbed and extinguished power by a collection of NPs in a solution, not necessarily equal. In addition, the corresponding single-particle property called absorption efficiency is defined as $\Phi_{\text{abs}} = \frac{\sigma_{\text{abs}}}{\sigma_{\text{ext}}}$. Where $\sigma_{\text{abs}}$ and $\sigma_{\text{ext}}$ are the single-particle absorption and extinction cross-sections.

To facilitate the comparison of the Au@Pd Core-shell studied in this manuscript with previous work, absorption efficiency $\Phi_{\text{abs}}(\lambda = 532\,\text{nm}) = \frac{\sigma_{\text{abs}}}{\sigma_{\text{ext}}}$ were calculated, in a water environment and no substrate (see Supplementary Fig. 22).

### Reporting summary

Further information on research design is available in the Nature Portfolio Reporting Summary linked to this article.

## Data availability

All data that support the findings of this study are available from the corresponding authors upon request. Source data are provided with this paper.

## Code availability

Software for data processing is available on GitHub and Zenodo[89].

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

## Acknowledgements
We thank Miguel Spuch Calvar for the graphics design. J.G. thanks Alina Ghisolfi and Silvia Zgryzek for their time dedicated to the care of his son Vicente, which made this manuscript possible. J.G. acknowledges the PRIME program of the German Academic Exchange Service (DAAD) with funds from the German Federal Ministry of Education and Research (BMBF). In addition, J.G. acknowledges the support from the Humboldt Foundation, the Royal Society of Chemistry (Research Fund R20-7244) and the Agencia Nacional de Promoción Científica y Tecnológica (PICT-2020-SERIEA-02966). A.S.-C. acknowledges Xunta de Galicia, Spain, for her postdoctoral fellowship. We acknowledge funding and support from the Deutsche Forschungsgemeinschaft (DFG, German Research Foundation) under Germany's Excellence Strategy, EXC 2089/1-390776260, the Bavarian program Solar Energies Go Hybrid (SolTech), the Center for NanoScience (CeNS), and the European Commission through the ERC Starting Grant CATALIGHT (802989).

## Author contributions
J.G. and E.C. designed the idea. M.H. and A.S.-C. performed the colloidal synthesis. M.H., A.S.-C., H.G., and R.G. characterized the colloids. J.G., I.L.V., and L.P.M. performed the optical measurements. J.G., M.B., and S.E. developed the thermal models. J.G., L.P.M., and M.B. processed and analyzed the data. J.G., I.L.V., S.S., S.A.M., F.D.S., and E.C. secured funding and supervised the project. M.H., J.G., S.E., I.L.V., L.P.M., A.S.-C., and M.B. made the Figures. J.G. wrote the original draft. All authors reviewed and edited the manuscript.

## Funding

## Competing interests
The authors declare no competing interests.
