## [Peer Review File · Nature Communications]

Reviewer comments, first round

Reviewer #1 (Remarks to the Author):

The authors use anti-Stokes thermometry to characterize the photothermal response of hybrid Pt-Au nanoparticles, and support measurements with theoretical models and numerical simulations. Both the experimental and numerical parts look properly conducted and everything is well explained.

However, I don't understand what makes this article relevant for a high-impact journal such as Nature Communication. AS thermometry is not a new temperature metrology technique. It has been introduced by the authors a couple of years ago, and the authors use it here the exact same way. The progress compared with the state of the art is that the authors are investigating Au-Pt nanoparticles and not only Au nanoparticles, and they mainly find that Pt nanodots are better than a Pt shell to favor photothermal conversion. It seems a bit incremental to warrant publication in Nature Communication for me. I reject this article for this reason, but I am not against the idea to publish this article in Nat Commun if the Editor thinks the message is suited for the Journal.

I just have a few concerns.

1- How do the authors estimate the average of Pt satellites (160) per NP from TEM images (Figure 4A). It seems very difficult to me, while a proper estimation is crucial to support the main message of the article. The authors should describe the exact procedure.

2- I did the calculation of the volumes of Pt in both cases, and don't find the same values.

In the satellite case:

$$V = N * \frac{4}{3} * \pi * r^3 = 160 * \frac{4}{3} * \pi * 2.5^3 = 1.05e4$$

and not 2.4e4.

In the shell case:

$$V = \frac{4}{3} * \pi * (r_{max}^3 - r_{min}^3) = \frac{4}{3} * \pi * ((30+1.8)^3 - 30^3)$$

2.16e4, not 1e4

Is there a mistake? If yes, and if my calculations are right, it would mean that the CS nanoparticles contain more Pt than the satellite-NP, which may explain the reason why they absorb less light. It would make the results much less interesting.

3- What is the dispersion of the sizes of the Pt satellites? In the TEM images, it looks large. Nevertheless, the authors are just using the averaged value of 5 nm. Be careful because the average volume of a population of spheres of radius $r=2.5$ nm is not $\frac{4}{3} * \pi * 2.5^3$, because of the power-of-3 non-linearity.

Reviewer #2 (Remarks to the Author):

The manuscript by J. Gargiulo et al. includes relevant measurements and a reasonable argumentation. It is noteworthy the demonstration that adding palladium in the form of satellites becomes beneficial (as compared with a core/shell structure) if the heating capability of the system needs to be maintained. It would be interesting to have a brief discussion on how this

structure would also influence the catalytic performance, which is the starting point of the manuscript (the authors have a work on the issue using related particles: Herran et al., *Adv. Funct. Mater.*, 2022, 32, 2203418. Also interesting on this regard is K. Sytwu et al., *Adv. Phys. X*, 2019, 4, 1619480). From an experimental point of view, it is significant the fact that heating abilities are studied using a thermometry approach measured on a single particle. The technique has been used before, but it is interesting here as the previous study on Au/Pd heating obtains data in solution measuring heating efficiency (I have found only one previous work on Au/Pd heating: Quintanilla et al., *Nanoscale* 2019, 11, 19561), and does not provide such a detailed thermal information.

The present manuscript, though, is lacking some discussion in the context of related works. Besides the mentioned discussion regarding catalysis, there are a couple of additional points. Their previous manuscript shows a blue-shift of the extinction of the core-shell nanoparticles. Another cited work shows a red shift (Forcherio et al., *ACS Nano*, 2022, 16, 12377). A blue shift is also predicted here through calculations for the absorption cross-section. Instead, a red shift of extinction is measured here when a Pd layer is grown, which the authors assign to the influence of CTAC. Since the calculation is only considering absorption and not scattering, how can they discard the effect of an increased scattering as the source of the shift in extinction? If we compare the absorption cross section and the measured scattering, their max does not overlap, and both affect the extinction. What's the reasoning of the shift in the other works? It is hard to compare the results here with the previous results mentioned in Au/Pd nanoparticles, as here a photothermal coefficient relating temperature and illumination intensity is used, while the previous work uses heating efficiency. However, some effort should be done on this direction to provide a general overview on the effect of Pd.

Along this line of thought, the provided photothermal coefficient is valid for the internal comparison of particles within the manuscript, as they all are measured in the same way. However, a theoretical question: can it be used to compare the structures studied here with other possible structures studied by other authors in terms of heating? I guess the answer is "yes" as long as only one particle is measured and heat dissipation conditions are maintained. From this perspective, if possible, it would be nice to offer a parameter that can be actually used to compare. Some other minor details should be addressed:

- In equation 2, the parameter a is not defined.
- Some references are lacking information (volume and page).
- There are no references in the supplementary information (though they are cited)
- How do the authors estimate the average number of palladium spheres in a core-satellite structure? The measure TEM is a 2D image, and they don't seem to be homogeneously distributed.
- What's the distance between the palladium spheres and the gold surface? This is a relevant parameter to define the plasmonic properties of the whole structure.
- Is the absorption cross-section linearly correlated with the photothermal coefficient?
- centrifuge speed should be given in units of g instead of rpm, so it can be really used in other laboratories (which may have a different centrifuge size).

Reviewer #3 (Remarks to the Author):

Plasmon-assisted catalysis proposes to make use of photogenerated hot carriers to help difficult chemical reactions. Potential applications of this process require a detailed knowledge of the respective roles of hot carriers and of photogenerated (photothermal) heating in activating the reaction. Therefore, precise estimates of the temperature of nanoparticles upon irradiation are required, which is exactly what this paper sets up to do.

The technique used is Anti-Stokes thermometry based on the photoluminescence spectrum of plasmonic gold nanoparticles (NPs). It is applied here to a quantitative study of the photothermal effect of gold nanospheres with catalyst metals Pt and Pd, as continuous shell or as "satellites", i.e., as much smaller NPs decorating the gold nanosphere.

Much of the paper is devoted to establishing a very solid basis for temperature estimates, using different methods, and checking the mutual consistency of these methods: anti-Stokes spectra, simulations, checks that particle structure is not modified by heating or by other processes such as the optical printing deposition. The paper produces very convincing estimates of photothermal

coefficients (i.e., the temperature elevation per unit laser intensity applied), which are in good agreement with previous published results, but with a higher accuracy.

The main original finding of the work is the significant difference in photothermal efficiency between a NP covered by a continuous shell of catalyst (here Pd) versus the same amount of Pd placed as satellite nanoparticles around the gold nanosphere. This observation can be attributed to enhanced damping of the gold plasmon by the continuous shell, whereas the satellites appear to hardly contribute any damping of the plasmon. Theoretical description by finite-element simulations and through an analytical solution of the heat conduction problem are in qualitative agreement with the experimental results. A more quantitative agreement may require questioning some of the approximations done.

The manuscript is clearly written and reads easily. My impression is of a high-quality scholarly contribution, with carefully performed experiments and simulations. Even minor effects such as the Kapitza resistance between materials, the explicit influence of the substrate and the potential perturbation due to the large electron temperature have been taken into account in the theoretical estimates. The final outcome of the study may not be very surprising, but it gives strong confidence in the quality of the measurements and the robustness of the temperature estimates they provide. The work is therefore of high quality, and will prove extremely useful to the researchers working on catalysis but also in biomedicine for photothermal therapy. It is less clear to me, however, how interesting this paper would be to the broader readership of Nat. Comm. and whether it could not appear in a more specialized journal. Irrespective of the editors' decision, I would advise the authors to revise their manuscript slightly considering the following comments.

Detailed comments

1. The manuscript mentions single-particle spectra in line 284, but displays "averaged" luminescence spectra in Fig. 4. It would be interesting to show single-particle spectra to convey a feeling for the noise and the dispersion of experimental results. Why is there a need for single-particle measurements if the results are ensemble-averaged in the end?
2. p. 337 and next ones: can the authors speculate on possible reasons of the increased damping in core-shell NPs? Is chemical interface damping enhanced by the larger contact area? Or do the authors assign plasmonic damping to bulk dissipation by Ohmic currents in the near-resonant shell of Pd? A lack of resonance in the case of Pd satellites could explain the unchanged damping.
3. It would be important to state in Table 1 that the photothermal coefficient is measured at the wavelength of 532 nm.
4. The paper is rather lengthy; the whole theory section could be shifted to SI.
5. Figure S4: The histograms Au NS 60 nm and Au60 Pd sat should have more bins! Same remark in Fig. S5 for Au67@Pd2 and Au67@Pd4.

Reviewer #4 (Remarks to the Author):

The authors describe their exceptionally thorough study of photothermal effects in Pd-coated Au nanoparticles. The authors combine calculations of optical absorption cross sections, state-of-the-art sample preparation using deterministic laser printing, and systematic measurements of temperature excursions as a function of laser fluence by electronic anti-Stokes Raman scattering. I highly recommend publication in Nature Communications after the authors have considered the following minor concerns:

- 1) The abstract uses the vague term "thermal performance". I encourage the authors to be more specific about what they mean. It seems to me that the results are ultimately simple: a change in the nanoparticle composition or structure changes the optical absorption and therefore changes the temperature excursion for a given fluence of laser excitation.

2) Similarly, at the end of the introduction on page 3, the authors state "...highlighting the role of the Au-Pd interfaces in the photothermal properties." If I understand the results correctly, the nature of the Pd coating (continuous film versus particles) changes the optical absorption. Nothing significant changes in the thermal properties.

3) I could not tell from the text what the substrate configuration is for the calculations of Figure 1D.

4) I encourage the authors to avoid normalized data if they can. For example, I do not see why the data for Figure 2C need to be normalized. That approach loses a lot of information. I understand that absolute units are difficult but since the authors are using printing of individual particles, the authors should be able to compare the scattering spectra without normalizing using the peak scattering.

5) On page 5, where the authors define $QY_{\{PL\}}^S$, they should state that the optical absorption is calculated (not measured).

6) In the middle of page 7, the reference to Figure 1B should be Figure 1D.

7) On page 9, "weekly" should be "weakly"

8) Similar to point #2 above, at the end of the text on page 9, the authors state "...the interface between the two metals is crucial and dictates the photothermal response..." I find this wording misleading. As stated in the next sentence, the effect of changing the nature of the Pd coating from is on the optical absorption. Nothing changes in the thermal behavior except that the nanoparticle absorbs more energy from the excitation laser. The Discussion section has a more examples of this problem at the bottom of page 10 and at the end of the second paragraph on page 11.

Reviewer #5 (Remarks to the Author):

The manuscript entitled 'Single particle thermometry in bimetallic plasmonic nanostructures' presents single-particle nanothermometry measurements. The authors investigated the link between morphology and thermal performance of colloidal Au/Pd nanoparticles with two different configurations: Au-Pd core-shell or satellites.

From my perspective, the authors elegantly selected the materials and prepared the high-uniform nanostructures. They employed hyperspectral AS thermometry to experimentally circumvent the challenge in photothermal modeling. The manuscript is well-written and finely-organized. In addition, data analysis and representation are of quite high quality. The objective of the study is good and the proposed concepts are characterized within the scope. However, the observations that authors have reported seemed to be highly confined to their system. Also, the fundamental mechanism behind the observed properties should be discussed. I recommend a major revision of the manuscript.

Major issues

1. The authors report certain photothermal properties of Au-Pd core-shell or satellites, which have relevance only for their specific nanostructure and materials. Other nanostructures with different materials, sizes, shapes, and different mediums would surely achieve different properties. Can the authors provide more general conclusions that would be widely applicable to other nanosystems, based on the observations in the current study? For example, for Au/Pd with different core/shell thicknesses, or for the bimetallic nanoparticles with other metals (Ag-Au, etc.)? The theoretical predictions could be presented in the conclusion part.
2. The fundamental mechanism behind the observed properties should be discussed. In Figure 2E,

the positive correlation between QY and Q should be discussed. In figure 4, why do the Pd satellites interact weakly with light?

3. The experimental process of AS has not been fully discussed. I wonder how the authors determine the relative position of the laser spot and the nanoparticle (Figure 3A)? If the laser keeps moving in a line, what is the spatial resolution of the spectroscopic measurements?

4. Continuing 3, is there any shape deformation of nanoparticles during laser irradiance? This will lead to variations in the relative position of NPs and the laser.

Minor issues:

1. Line 288: 'weekly' should be 'weakly'

2. In SM, line 283, "S11. Optical properties of Pd satellites", should be erased.

3. The authors used the seed-mediated method to form nanoparticles and CTAC as stabilizers. Usually, the shape of Au NPs via this method would be close to an octahedron, since CTAC prefers to attach to {111} facet. Would that affect the photothermal effect at different orientations? Did the authors apply NP aging to make smooth Au spheres?

Reviewer #6 (Remarks to the Author):

Studying plasmonic metals and plasmon-assisted catalysis is of far-reaching interest to the general scientific community. The authors reported the growth of colloidal Au/Pd nanoparticles with different core/shell structures and analyzed the impact of morphology and thickness on the photothermal properties of such hybrid structures. The authors have demonstrated that AS thermometry can provide information at a single particle level in a non-destructive way, which may help the understanding of other complex structure interfaces. Solid and insightful discussions are provided and I quite liked reading them. However, my main concerns reside in the sample quality and data analysis shown at the beginning of the manuscript.

The authors synthesized several Au/Pd core/shell structures, and used electron microscopy and/or EDS to identify their thickness and morphology. A very accurate thickness of the Pd shell is provided. For example, in line 127, the authors claim the Pd thickness are 2.4 ± 0.3 nm, and 3.6 ± 0.3 nm. My questions are how these numbers are calculated. The EDS shown in Fig. 1b looks very noisy, and the distribution of the Pd shell is obviously inhomogeneous. The authors should provide a detailed calculation process to repeat these results. In fig. 4a, the Pd shells all melt together and thickness is again apparently inhomogeneous. It is quite surprising that a 0.3 nm error bar can be realized. In addition, many Au cores look not well faceted or irregular, but a 0.1 nm or 0.2 nm error bar is still given for the median size.

One interesting conclusion in this paper is that the photothermal coefficient is relatively irrelevant to the Pd shell thickness. Given the irregular Pd shell, I highly suggest the authors revisit the data and exclude the possibility of sample bias. In addition, the structural information of the Au/Pd interface is relatively unclear. Pd is grown epitaxially on Au or what is the crystallographic information along the interface region.

Overall, this is a nice paper that contains many high-quality experimental results. The discussion is relatively clear. However, careful calculations and insightful discussions are suggested to allow the manuscript to be published in a high-profile journal like Nat. Commun.

There are some minor grammatical errors and although none interfere with understanding and reading. Careful proofreading is recommended. Here are some minor typos or errors listed below:

Line 255, change constant to constants

Line 369, remove additional space before the comma

Line 372, remove additional space before the comma, add in one space before ammonium hydroxide.

Response to reviewers:

We would like to thank all the reviewers for the constructive comments, which have motivated the improvement of our manuscript. Here we provide a point-by-point response to each comment, **in bold font**, indicating when opportune the modifications made to the manuscript (in blue).

Reviewer #1:

The authors use anti-Stokes thermometry to characterize the photothermal response of hybrid Pt-Au nanoparticles, and support measurements with theoretical models and numerical simulations.

Both the experimental and numerical parts look properly conducted and everything is well explained.

We thank Reviewer 1 for the positive appraisal of our work.

However, I don't understand what makes this article relevant for a high-impact journal such as Nature Communication. AS thermometry is not a new temperature metrology technique. It has been introduced by the authors a couple of years ago, and the authors use it here the exact same way. The progress compared with the state of the art is that the authors are investigating Au-Pt nanoparticles and not only Au nanoparticles, and they mainly find that Pt nanodots are better than a Pt shell to favor photothermal conversion. It seems a bit incremental to warrant publication in Nature Communication for me.

We would like to remark the novel aspects of our work and its significance. Temperature is a key parameter in plasmon-assisted catalysis. However, most of the previous experimental work studying light-to-heat conversion of plasmonic-catalytic nanoparticles were performed at the bulk level of colloidal solutions. In this manuscript, we show that AS thermometry provides direct information on the temperature at the nanoparticle surface, which can strongly differ from the bulk temperature of the solution and is the most relevant for heterogenous catalysis.

In addition, it must be noted that Anti-Stokes Nanothermometry is a very novel temperature metrology technique. Until now, the technique has only been employed in monometallic Au nanoparticles. Here, we have demonstrated that its use can be extended to bimetallic nanocomposites. We believe that proving that a non-destructive method, compatible with in operando conditions such as Anti-Stokes Nanothermometry can be applied to such complex materials is of great interest to the broad nanoscience community and exceeds the context of plasmonic chemistry.

I reject this article for this reason, but I am not against the idea to publish this article in Nat Commun if the Editor thinks the message is suited for the Journal.

I just have a few concerns.

1- How do the authors estimate the average of Pt satellites (160) per NP from TEM images (Figure 4A). It seems very difficult to me, while a proper estimation is crucial to support the main message of the article. The authors should describe the exact procedure.

We have performed a more precise characterization of the Au60-Pd-satallites system.

The number of satellites was estimated in two ways: 1) Using TEM images and 2) Using Inductive Coupled Plasma Atomic Emission Spectroscopy (ICP-AES).

1. Number of satellites using TEM images.

Figure S18a shows the histogram of the number of visible satellites $N_{visible}$ per Au core, estimated by counting satellites on the TEM images. The mean value is $\overline{N_{visible}} = (27 \pm 1)$ with a standard deviation of $\sigma = 6$.

However, TEM images are 2D, so some satellites will not be visible. Only satellites close to the equator of the Au core will be visible. On the contrary, if the position of the satellite is below or above the Au core, it will be masked by the Au core. Figure S18b illustrates this situation. It is considered that satellites in positions marked with a #1 will be visible, #2 will be partially visible and #3 and #4 will not be visible. The blue-shaded angle correspond to the solid angle of the sphere where satellites are visible or partially visible. This angle can be estimated as $\cos \alpha_T = \frac{r_C}{r_C + D_S}$, where r_C is the radius of the core and D_S is the diameter of the satellite.

The fraction F of the surface area that is visible can then be calculated as a surface integral:

$$F = \frac{\int_0^{2\pi} \int_{\frac{\pi}{2}-\alpha_T}^{\frac{\pi}{2}+\alpha_T} r^2 \sin \theta d\theta d\varphi}{4\pi r^2} = \frac{1}{2} \int_{\frac{\pi}{2}-\alpha_T}^{\frac{\pi}{2}+\alpha_T} \sin \theta d\theta = \sin \alpha_T$$

Using $r_C = (29.9 \pm 0.2)$ nm and $D_S = (6.3 \pm 0.2)$ nm, $\alpha_T \approx 34,3^\circ$ this yields $F = (0.56 \pm 0.03)$.

This means that around 56% of the satellites will be visible in the 2D TEM images.

The average number of satellites N_s can be calculated as $N_s = \frac{N_{visible}}{F} = (48 \pm 4)$.

Figure S19. Number of satellites using TEM images. (a) Histogram of the number of visible satellites $N_{visible}$. The mean value is $\overline{N_{visible}} = (27 \pm 1)$ with a standard deviation of $\sigma = 6$. (b) Illustration of the TEM imaging. The large sphere represents the Au core, and the small red spheres represent the Pd satellites. The satellites that are between the two vertical dashed lines are not visible in the TEM images. The blue shaded area corresponds to the solid angle of the sphere that has visible satellites.

2. Number of satellites using ICP-AES.

Alternative estimation of the number of satellites was using (ICP-AES). This technique allows the quantification of the ratio between the Au and Pd mass. The measured value was $\frac{M_{Pd}}{M_{Au}} = (0.036 \pm 0.002)$, where M_{Pd} is the added mass of all the Pd satellites and M_{Au} is the mass of the Au core. The total volume of palladium per NP is $N_s V_{sat} = V_C \frac{\rho_{Au} M_{Pd}}{\rho_{Pd} M_{Au}}$, with V_C the volume of a Au core, V_{sat} the volume of a Pd satellite and N_s the number of satellites.

Assuming spherical shape of Au core with a diameter of (59.7 ± 0.2) nm and $V_{sat} = (120 \pm 10)$ nm³, $N_s = (54 \pm 7)$.

Both methods provide similar values for the number of satellites.

The average value between both techniques is $N_s = (51 \pm 6)$. It must be noted that this number is significantly lower than the one informed in the previous version of the manuscript. The reason for this is that we had incorrectly modelled the factor F , leading to the wrong value of $F = 0.17$.

This discussion was included in the new section S11 of the revised Supplementary Materials, called *Characterization of Au60-Pd-sat NPs*. In the main manuscript, the following sentence was included.

“Detailed characterizations of the of the three systems are provided in sections S1 and S11 of the SM.

The average number of satellites per NP of the Au60-Pd-sat was estimated to be $\langle N_s \rangle = (51 \pm 6)$.”

2- I did the calculation of the volumes of Pt in both cases, and don't find the same values.

In the satellite case:

$$V = N * \frac{4}{3} * \pi * r^3 = 160 * \frac{4}{3} * \pi * 2.5^3 = 1.05e4$$

and not 2.4e4.

In the shell case:

$$V = \frac{4}{3} * \pi * (r_{\text{max}}^3 - r_{\text{min}}^3) = \frac{4}{3} * \pi * ((30+1.8)^3 - 30^3)$$

2.16e4, not 1e4. Is there a mistake? If yes, and if my calculations are right, it would mean that the CS nanoparticles contain more Pt than the satellite-NP, which may explain the reason why they absorb less light. It would make the results much less interesting.

We have performed a more precise characterization of the Pd to Au mass ratios using ICP-AES. The revised version of the manuscript now reads:

ICP-AES was employed to quantify the Pd to Au mass ratio $\frac{M_{Pd}}{M_{Au}}$ of each system, yielding the values $\frac{M_{Pd}}{M_{Au}} = (0.073 \pm 0.004)$ and $\frac{M_{Pd}}{M_{Au}} = (0.036 \pm 0.002)$, for Au60@Pd2 and Au60-Pd-sat respectively. This means that the amount of Pd is on the same order of magnitude for both configurations, being the total mass of Pd in the satellites approximately half the mass in a 2 nm Pd shell.

The reason why Au60@Pd2 absorb significantly less light than the Au60-Pd-sat is the higher plasmon damping due to the larger contact area between Au and Pd. We have added the following sentences to highlight this fact:

While the Q-factor of the Au60@Pd2 system is reduced by ~27% in comparison with the Au cores, it is only reduced by ~3% for the Au60-Pd-sat. This can be explained by the significantly lower contact area between Au and Pd in the Au60-Pd-sat system in comparison with the Au60@Pd2 (see section S12 of the SM for further details).

Overall, it can be concluded that the inclusion of Pd affects the photothermal coefficient of Au NPs mainly by altering their absorption cross-section due to plasmon damping. The main characteristic governing the damping is the presence of an interface between materials, with a minor influence of the total amount of added Pd. In this sense, the experiments presented here highlight the importance of the interface management for light to heat conversion and provide guidelines on how to design future nanocomposites for the maximization (or minimization) of their photothermal response.

3- What is the dispersion of the sizes of the Pt satellites? In the TEM images, it looks large. Nevertheless, the authors are just using the averaged value of 5 nm. Be careful because the average volume of a population of spheres of radius $r=2.5$ nm is not $\frac{4}{3} * \pi * 2.5^3$, because of the power-of-3 non-linearity.

We thank the reviewer for raising this point. We have performed a more careful size characterization of the Pd satellites using TEM microscopy. We have included the following figure as Figure S15 in the revised Supplementary Material. The mean diameter is $d = 6.3$ nm with a standard deviation of $\sigma_d = 1.3$ nm. The mean volume of a single Pd satellite was estimated to be $V = 120$ nm³ with a standard deviation of $\sigma_V = 100$ nm³. Finally, we invite Reviewer #1 to read the revised section S11 of the SM, that contains a more comprehensive characterization of the Au₆₀-Pd-sat NPs.

Figure S15. Size characterization of Pd satellites. (a) Histogram of the diameter of Pd Satellites. The mean diameter is $d = 6.3$ nm with a standard deviation of $\sigma_d = 1.3$ nm. (b) Histogram of the diameter of Pd Satellites. The mean volume of a single Pd satellite was estimated to be $V = 120$ nm³ with a standard deviation of $\sigma_V = 100$ nm³. (c) TEM images of the Pd satellites.

Reviewer #2:

The manuscript by J. Gargiulo et al. includes relevant measurements and a reasonable argumentation. It is noteworthy the demonstration that adding palladium in the form of satellites becomes beneficial (as compared with a core/shell structure) if the heating capability of the system needs to be maintained. It would be interesting to have a brief discussion on how this structure would also influence the catalytic performance, which is the starting point of the manuscript (the authors have a work on the issue using related particles: Herran et al., *Adv. Funt. Mater*, 2022, 32, 2203418. Also interesting on this regard is K. Sytwu et al., *Adv. Phys. X*, 2019,4, 1619480).

We thank the reviewer for the suggestion, and we have included the suggested references to the manuscript:

For example, in prior studies by Herran *et al.*, it was demonstrated that core-satellite systems show a higher enhancement for hydrogen generation upon illumination, compared to core-shell ones.

Our work presents information that can contribute to the design rule of plasmonic bimetallic photocatalysts. In our findings, we determined experimentally a similar photothermal coefficient for 60 nm bare Au nanospheres (AuNS) and the same core surrounded by Pd satellites (Au60-Pd-sat). So we concluded that the absence of an interface between the core and satellites is crucial to keep the photothermal behavior unaffected. Therefore, in this type of catalysts, the reactor not only benefit from their ability to create hotspots (as described by Herran *et al.*, but also from the energy funneled in the form of heat. Thus, taking advantage of both ways that the input of energy can be dissipated upon LSPR excitation to drive chemical reactions.

From an experimental point of view, it is significant the fact that heating abilities are studied using a thermometry approach measured on a single particle. The technique has been used before, but it is interesting here as the previous study on Au/Pd heating obtains data in solution measuring heating efficiency (I have found only one previous work on Au/Pd heating: Quintanilla et al., *Nanoscale* 2019, 11, 19561), and does not provide such a detailed thermal information.

We thank the reviewer for the appraisal and the reference, which we have included in the manuscript. We have also included the following sentence:

Using AS thermometry, it was possible to determine the temperature increases at the surface of the Au-Pd NPs, in opposition to previous studies focused on the bulk heating of solutions.

The present manuscript, though, is lacking some discussion in the context of related works. Besides the mentioned discussion regarding catalysis, there are a couple of additional points. Their previous manuscript shows a blue-shift of the extinction of the core-shell nanoparticles. Another cited work shows a red shift (Forcherio et al., *ACS Nano*, 2022, 16, 12377). A blue shift is also predicted here through calculations for the absorption cross-section. Instead, a red shift of extinction is measured here when a Pd layer is grown, which the authors assign to the influence of CTAC. Since the calculation is only considering absorption and not scattering, how can they discard the effect of an increased scattering as the source of the shift in extinction? If we compare the absorption cross section and the measured scattering, their max does not overlap, and both affect the extinction. What's the reasoning of the shift in the other works?

The revised Supplementary Materials has a new Figure S3. It now includes the calculation of scattering and extinction cross sections, in addition to absorption. It must be noted that for Au@Pd Core Shell NPs, all three calculated spectra (absorption, scattering and extinction) present a blue shift with respect to the bare Au NP.

For this reason, we believe that the large concentration of CTAC (20 mM) used as the stabilizing agent of the CS-NPs is the source of the observed red shift, as reported by Movsesyan and co-workers (J. Opt. Soc. Am. A 36, C78-C84 (2019)). This idea is supported by the fact that no significant redshift was observed for the Au60@Pd2 NPs, as shown in Table 1 of the manuscript. In the last step of the synthesis, the Au60@Pd2 NPs were redispersed in ultrapure H₂O instead of CTAC.

Fig. S3. Au67@Pd absorption, scattering and extinction cross sections (m²) for different interfaces between the NP and the glass substrate.

It is hard to compare the results here with the previous results mentioned in Au/Pd nanoparticles, as here a photothermal coefficient relating temperature and illumination intensity is used, while the previous work uses heating efficiency. However, some effort should be done on this direction to provide a general overview on the effect of Pd.

Along this line of thought, the provided photothermal coefficient is valid for the internal comparison of particles within the manuscript, as they all are measured in the same way. However, a theoretical question: can it be used to compare the structures studied here with other possible structures studied by other authors in terms of heating? I guess the answer is “yes” as long as only one particle is measured and heat dissipation conditions are maintained. From this perspective, if possible, it would be nice to offer a parameter that can be actually used to compare.

The heating efficiency or photothermal efficiency η is usually employed in the context of illuminated NPs solutions. In that context, heating is a collective effect of many NPs suspended in a solvent. The relevant parameter to describe such a system is the bulk temperature increase of the solution ΔT_s . In the steady state, it can be calculated by (see, for example J. Phys. Chem. C 2013, 117, 51, 27073–27080)

$$\Delta T_s = \frac{Q_{in}}{m_s C_s B} \quad (A)$$

where m_s and C_s are the mass and the heat capacity of the solvent, Q_{in} is the total heat dissipated from the NPs into the solution, and B is the rate of heat dissipation from the solution to the external environment. It must be noted that the bulk temperature increase of the solution ΔT_s is different (usually, much lower) than the surface temperature increase of the NPs suspended ΔT_{NP} in the solution.

Typically, Q_{in} is calculated as

$$Q_{in} = (P_0 - P_T)\eta = P_0(1 - 10^{-OD})\eta \quad (B)$$

where P_0 and P_T are the incident and transmitted powers, η is the *photothermal efficiency* of the solution, and OD is the optical density of the solution. This parameter is defined as the ratio between the total absorbed and extinguished power by a collection of NPs in a solution, not necessarily equal. In addition, the corresponding single-particle property *absorption efficiency* is defined as $\Phi_{abs} = \frac{\sigma_{abs}}{\sigma_{ext}}$. Where σ_{abs} and σ_{ext} are the single-particle absorption and extinction cross-sections.

In summary, the *photothermal efficiency* η is an ensemble property that tells which fraction of the extinguished power is absorbed. The *absorption efficiency* Φ_{abs} is its single-particle counterpart. When the focus of interest is to heat solutions, the total released heat is the magnitude of interest, and these efficiencies become relevant. On the other hand, β is a single-particle property that directly links the incident irradiance with the particle temperature increase $\Delta T_{NP} = \beta I_{exc}$. β is proportional to the absorption of light but is also affected by the heat dissipation properties of the environment. It becomes relevant when the focus is the temperature of the particle ΔT_{NP} , and not the bulk solution ΔT_s .

Therefore, they are different properties and used in different contexts. One links absorption with extinction, the other links irradiance with temperature increase. It must be noted that a NP with a higher *photothermal efficiency* η not necessarily has a higher photothermal coefficient β . Consider for example the case of two Au Nanospheres of 10 and 100 nm. While the smallest will have a higher *photothermal efficiency*, the largest will have a higher photothermal coefficient.

Nevertheless, in order to facilitate the comparison of our systems with previous works, we have computed the *absorption efficiency* Φ_{abs} of the Au@Pd Core-shell NPs. We have included a new section in the revised SM, that includes the following figure:

Figure S23. Calculated Absorption Efficiencies at 532 nm. Green and blue dots correspond to Au@Pd Core-shell NPs with a core size of 60 nm and 67 nm respectively.

Some other minor details should be addressed:

- In equation 2, the parameter a is not defined.

We thank the reviewer 2 for this observation. The following sentence was included to define it: the temperature \$T^{NP}\$ of a spherical NP of radius \$a\$

- Some references are lacking information (volume and page).

Thank you. We have corrected them.

- There are no references in the supplementary information (though they are cited).

We have included all the references in the supplementary materials.

- How do the authors estimate the average number of palladium spheres in a core-satellite structure? The measure TEM is a 2D image, and they don't seem to be homogeneously distributed.

Please see answer to question 1 from reviewer #1.

- What's the distance between the palladium spheres and the gold surface? This is a relevant parameter to define the plasmonic properties of the whole structure.

We thank the reviewer for this important question. In order to discuss the answer, the revised version of the supplementary materials includes the following new section:

S12. The gap size between the palladium satellites and the gold surface.

The Au60-Pd-sat system is produced by solvent-induced electrostatic self-assembly between a positive CTAC capped Au NP and negative Polyvinylpyrrolidone (PVP) capped Pd satellites. Therefore, the gap between both surfaces is expected to be in the order of magnitude of the size of the capping molecules.

To confirm this, we have performed STEM-HAADF, as shown in Figure S21. Due to the 3D geometry of the system, the satellites that are above or below the equatorial plane of the Au sphere appear like they are penetrating the Au core. For this reason, the gap was measured in the satellites that look further from the Au surface. The distance between surfaces is approximately 1 nm, which can be considered to be an upper bound. On the other hand, the fact that we observed a gap for the NPs in the equatorial plane suggest that such a gap exists in most cases, and that the contact surface between materials is negligible.

Figure S21. Determination of gap sizes. (a) Scanning Transmission Electron Microscopy of a Au60-Pd satellites NP. The two profiles drawn in the figure are plot in (b).

To further discuss the impact of the gap size in the optical properties, we have calculated the absorption cross section of the system for different gap distances between the Au and the Pd satellites, as shown in Figure S22. The change in absorption at 532 nm with respect to an Au core without satellites is -5.5%, -4%, and -0.5% for gap sizes of 0 nm, 0.5 nm, and 1 nm, respectively. These estimated reductions are consistent with the experimentally measured 2% reduction of the mean photothermal coefficient β between AuNS-60nm and Au60-Pd-sat.

Figure S22. Absorption cross section of a Au60-Pd satellites for different gap sizes between the satellites and the Au surface. The green line indicates the 532 nm laser.

In addition,

Reviewer #3:

Plasmon-assisted catalysis proposes to make use of photogenerated hot carriers to help difficult chemical reactions. Potential applications of this process require a detailed knowledge of the respective roles of hot carriers and of photogenerated (photothermal) heating in activating the reaction. Therefore, precise estimates of the temperature of nanoparticles upon irradiation are required, which is exactly what this paper sets up to do.

The technique used is Anti-Stokes thermometry based on the photoluminescence spectrum of plasmonic gold nanoparticles (NPs). It is applied here to a quantitative study of the photothermal effect of gold nanospheres with catalyst metals Pt and Pd, as continuous shell or as “satellites”, i.e., as much smaller NPs decorating the gold nanosphere.

Much of the paper is devoted to establishing a very solid basis for temperature estimates, using different methods, and checking the mutual consistency of these methods: anti-Stokes spectra, simulations, checks that particle structure is not modified by heating or by other processes such as the optical printing deposition. The paper produces very convincing estimates of photothermal coefficients (i.e., the temperature elevation per unit laser intensity applied), which are in good agreement with previous published results, but with a higher accuracy.

The main original finding of the work is the significant difference in photothermal efficiency between a NP covered by a continuous shell of catalyst (here Pd) versus the same amount of Pd placed as satellite nanoparticles around the gold nanosphere. This observation can be attributed to enhanced damping of the gold plasmon by the continuous shell, whereas the satellites appear to hardly contribute any damping of the plasmon. Theoretical description by finite-element simulations and through an analytical solution of the heat conduction problem are in qualitative agreement with the

experimental results. A more quantitative agreement may require questioning some of the approximations done.

The manuscript is clearly written and reads easily. My impression is of a high-quality scholarly contribution, with carefully performed experiments and simulations. Even minor effects such as the Kapitza resistance between materials, the explicit influence of the substrate and the potential perturbation due to the large electron temperature have been taken into account in the theoretical estimates. The final outcome of the study may not be very surprising, but it gives strong confidence in the quality of the measurements and the robustness of the temperature estimates they provide. The work is therefore of high quality, and will prove extremely useful to the researchers working on catalysis but also in biomedicine for photothermal therapy. It is less clear to me, however, how interesting this paper would be to the broader readership of Nat. Comm. and whether it could not appear in a more specialized journal. Irrespective of the editors' decision, I would advise the authors to revise their manuscript slightly considering the following comments.

We thank the reviewer for the assessment and the positive appraisal of the work.

Detailed comments

1. The manuscript mentions single-particle spectra in line 284, but displays “averaged” luminescence spectra in Fig. 4. It would be interesting to show single-particle spectra to convey a feeling for the noise and the dispersion of experimental results. Why is there a need for single-particle measurements if the results are ensemble-averaged in the end?

We have included a new Figure S8 in the revised Supplementary Materials that shows single-particle luminescence spectra for all the systems studied in the manuscript. In addition, information on the dispersion of experimental results is provided in the histograms of Figures 1,3, and 4 of the main manuscript and Figures S5, S6, and S7 of the revised Supplementary Materials. However, we opted to show ensemble-averaged in Figures 2 and 4 of the main text because it facilitates the physical discussion.

Figure S8. Single particle PL emission spectra, excited with $1 \text{ mW}/\mu\text{m}^2$ of laser light at 532 nm. Band with no data corresponds to the laser rejection filter.

2. p. 337 and next ones: can the authors speculate on possible reasons of the increased damping in core-shell NPs? Is chemical interface damping enhanced by the larger contact area? Or do the authors

assign plasmonic damping to bulk dissipation by Ohmic currents in the near-resonant shell of Pd? A lack of resonance in the case of Pd satellites could explain the unchanged damping.

As the reviewer suggests, increased damping could be due to surface effects (increased electron-surface scattering, plasmon decay into interfacial states) or bulk effects (ohmic damping in the Pd shell). It must be noted that an increase in linewidth is an indication of plasmonic damping, but that the underlying mechanism cannot be determined only by studying linewidth changes. Nevertheless, the lower damping for the core-satellites in comparison with the core-shell would indicate that the surface mechanisms dominate, in line with the reviewer's speculation.

A complete description of the mechanistic details for plasmonic damping upon Pd inclusion is out of the scope of the manuscript. However, we have included the following sentence together with three new references for the readers that are interested in deepening into the topic.

Increased damping could be due to surface effects (increased electron-surface scattering, plasmon decay into interfacial states) or bulk effects (ohmic damping in the Pd shell).

70. Zheng, Z., Tachikawa, T. & Majima, T. Single-particle study of Pt-modified Au nanorods for plasmon-enhanced hydrogen generation in visible to near-infrared region. *J. Am. Chem. Soc.* 136, 6870–6873 (2014).

71. Joplin, A. et al. Correlated Absorption and Scattering Spectroscopy of Individual Platinum-Decorated Gold Nanorods Reveals Strong Excitation Enhancement in the Nonplasmonic Metal. *ACS Nano* 11, 12346–12357 (2017).

72. Zijlstra, P., Paulo, P. M. R., Yu, K., Xu, Q.-H. & Orrit, M. Chemical Interface Damping in Single Gold Nanorods and Its Near Elimination by Tip-Specific Functionalization. *Angew. Chemie Int. Ed.* 51, 8352–8355 (2012).

3. It would be important to state in Table 1 that the photothermal coefficient is measured at the wavelength of 532 nm.

The reviewer is right. Both the heading and the caption of the table now states that it was measured at 532 nm.

4. The paper is rather lengthy; the whole theory section could be shifted to SI.

We thank the reviewer for the suggestion. Most of the theory is already in the Materials and Methods section or in the Supplementary Materials. We believe that further reducing the length of the theory section in the main text might compromise its clarity.

5. Figure S4: The histograms Au NS 60 nm and Au60 Pd sat should have more bins! Same remark in Fig. S5 for Au67@Pd2 and Au67@Pd4.

We thank the reviewer for the suggestion. It is true that the mentioned histograms should have more bins if they were presented isolated. However, we decided to keep the same bin sizes and number of bins for all the histograms of the same figure, to facilitate the comparison between systems.

Reviewer #4:

The authors describe their exceptionally thorough study of photothermal effects in Pd-coated Au nanoparticles. The authors combine calculations of optical absorption cross sections, state-of-the-art sample preparation using deterministic laser printing, and systematic measurements of temperature excursions as a function of laser fluence by electronic anti-Stokes Raman scattering. I highly

recommend publication in Nature Communications after the authors have considered the following minor concerns:

We thank the reviewer for the positive appraisal of the work.

1) The abstract uses the vague term “thermal performance”. I encourage the authors to be more specific about what the mean.

The terms *photothermal material* or *photothermal effect* are usually employed to mean “the temperature increase of a material due to the absorption of light”. (See for example <https://doi.org/10.1039/9781839165177-00001>). To improve the clarity of the text, we have performed the following changes:

- The term “thermal performance” was changed to “light-to-heat conversion” in the Abstract
- In the introduction, the phrase: “First, Au@Pd CS-NPs with different shell thicknesses are designed to investigate changes in their on-resonance photothermal response.” Was expanded to “First, Au@Pd CS-NPs with different shell thicknesses are designed to investigate changes in their on-resonance photothermal response, understood as the NPs’s temperature increase under illumination.”

It seems to me that the results are ultimately simple: a change in the nanoparticle composition or structure changes the optical absorption and therefore changes the temperature excursion for a given fluence of laser excitation.

2) Similarly, at the end of the introduction on page 3, the authors state “...highlighting the role of the Au-Pd interfaces in the photothermal properties.” If I understand the results correctly, the nature of the Pd coating (continuous film versus particles) changes the optical absorption. Nothing significant changes in the thermal properties.

The property measured in this manuscript is the photothermal coefficient β that links the temperature increase of a NP with the excitation irradiance through the relation $T^{NP} = \beta I_{exc} + T_0$. The photothermal coefficient of an object depends on two phenomena: i) Its ability to produce heat upon light absorption and ii) Its ability to dissipate the heat to its environment. A change in the morphology of the object can in principle affect both. However, our experimental technique measures β , without the possibility to distinguish between the independent contribution of absorption and conduction. That is why we kept the discussion in terms of photothermal response.

The reviewer is right when saying that for the NPs presented in our manuscript, the observed changes in the photothermal coefficients can be explained mainly by changes in the absorption, with a minor contribution from changes in the heat conduction. However, this is a conclusion of the manuscript, that is known only after discussing the experimental results and comparing with the calculations of absorption cross sections.

For clarity, the phrase: “It is found that the generation of heat is larger if the Pd is included as satellites around the Au core than if it is included as a shell, highlighting the role of the Au-Pd interfaces in the photothermal properties.” Was changed to

It is found that the absorption of light is larger if the Pd is included as satellites around the Au core than if it is included as a shell, highlighting the role of the Au-Pd interfaces in the photothermal response.

3) I could not tell from the text what the substrate configuration is for the calculations of Figure 1D.

We thank the reviewer for finding this error. It now reads:

Figure 1D shows the numerically calculated absorption cross section spectra for different Pd thicknesses, in a homogeneous water environment.

4) I encourage the authors to avoid normalized data if they can. For example, I do not see why the data for Figure 2C need to be normalized. That approach loses a lot of information. I understand that absolute units are difficult but since the authors are using printing of individual particles, the authors should be able to compare the scattering spectra without normalizing using the peak scattering.

The main message to transmit in Figure 2C is the redshift and broadening of the scattering spectra upon inclusion of the Pd shell. The best way to transmit that message is by using normalized data. On the other hand, without the presence of a scattering standard, day to day variations in the illumination could cause misleading conclusions if the data is not normalized. It must be noted that information on the dispersion of experimental results is given in the histograms of Figures S5 and S6 of the revised Supplementary Materials. In addition, the new Figure S3 of the Supplementary Materials shows the calculated scattering cross section for the CS NPs of Figure 2, without significant variations of the peak intensity.

On the contrary, we are showing photoluminescence data without normalization in Figure 2D. In this case, we explicitly intend to show the significant changes in emission intensity upon Pd inclusion.

5) On page 5, where the authors define $QY_{\{PL\}}^S$, they should state that the optical absorption is calculated (not measured).

We have included the following sentence to make it clear: The value of $\sigma_{abs}(\lambda = 532nm)$ was simulated for the different NPs on glass substrates and water environments, as described in section S2 of the SM.

6) In the middle of page 7, the reference to Figure 1B should be Figure 1D.

We thank the reviewer; it was changed accordingly.

7) On page 9, “weekly” should be “weakly”

We thank the reviewer; the typo was corrected.

8) Similar to point #2 above, at the end of the text on page 9, the authors state “...the interface between the two metals is crucial and dictates the photothermal response...” I find this wording misleading. As stated in the next sentence, the effect of changing the nature of the Pd coating from is on the optical absorption. Nothing changes in the thermal behavior except that the nanoparticle absorbs more energy from the excitation laser. The Discussion section has a more examples of this problem at the bottom of page 10 and at the end of the second paragraph on page 11.

Please see the response to point #2 above. The revised version of the manuscript now reads:

Overall, it can be concluded that the inclusion of Pd affects the photothermal coefficient of Au NPs mainly by altering their absorption cross-section due to plasmon damping. The main characteristic governing the damping is the presence of an interface between materials, with a minor influence of the total amount of added Pd. In this sense, the experiments presented here highlight the importance of the interface management for light to heat conversion and provide guidelines on how to design future nanocomposites for the maximization (or minimization) of their photothermal response.

Reviewer #5:

The manuscript entitled 'Single particle thermometry in bimetallic plasmonic nanostructures' presents single-particle nanothermometry measurements. The authors investigated the link between morphology and thermal performance of colloidal Au/Pd nanoparticles with two different configurations: Au-Pd core-shell or satellites.

From my perspective, the authors elegantly selected the materials and prepared the high-uniform nanostructures. They employed hyperspectral AS thermometry to experimentally circumvent the challenge in photothermal modeling. The manuscript is well-written and finely-organized. In addition, data analysis and representation are of quite high quality. The objective of the study is good and the proposed concepts are characterized within the scope. However, the observations that authors have reported seemed to be highly confined to their system. Also, the fundamental mechanism behind the observed properties should be discussed. I recommend a major revision of the manuscript.

We thank the reviewer for the positive appraisal of our work.

Major issues:

1. The authors report certain photothermal properties of Au-Pd core-shell or satellites, which have relevance only for their specific nanostructure and materials. Other nanostructures with different materials, sizes, shapes, and different mediums would surely achieve different properties. Can the authors provide more general conclusions that would be widely applicable to other nanosystems, based on the observations in the current study? For example, for Au/Pd with different core/shell thicknesses, or for the bimetallic nanoparticles with other metals (Ag-Au, etc.)? The theoretical predictions could be presented in the conclusion part.

The reviewer is right, the photothermal coefficient of NPs is dependent on the material, size, shape and medium. In this manuscript, we have demonstrated that Anti-Stokes Nanothermometry is applicable to bimetallic NPs. We believe that proving that a non-destructive, compatible with in operando conditions such as Anti-Stokes Nanothermometry can be applied to such complex materials is of great interest to the broad nanoscience community and exceeds the context of plasmonic chemistry. The results pave the way for the future use of Anti-Stokes Nanothermometry to characterize other NPs of interest.

In addition, we believe that there are general conclusions that are applicable to other bimetallic nanostructures:

For bimetallic core shells such as Au-Pd, because both materials are highly conductive, adding Pd does not significantly restrict the conduction of heat, meaning that the main parameter influencing their photothermal coefficients is the absorption cross-section. Figure S24 of section S14 of the revised Supplementary Materials shows the calculated absorption cross sections for CS NPs with an Au core and shells of different materials, including Ag, Pt and Rh.

Figure S24. Prediction of median temperature increase for Au@M core-shell structures (M=Pd, Pt, Rh, Ag). (a) Absorption cross-section (σ_{Abs}) for Au@M core-shell nanoparticles across the visible range. 60 nm Au NS were used as core, while the shell was 2 nm thick. (b) Predicted temperature increase for all systems as a function of wavelength, corresponding to an irradiance of $1 \text{ mW}/\mu\text{m}^2$.

Another conclusion is that the presence of an interface between metals has a higher impact on the absorption cross section than the total amount of added Pd. This is now highlighted in the following sentence of the conclusion:

Overall, it can be concluded that the inclusion of Pd affects the photothermal coefficient of Au NPs mainly by altering their absorption cross-section due to plasmon damping. The main characteristic governing the damping is the presence of an interface between materials, with a minor influence of the total amount of added Pd. In this sense, the experiments presented here highlight the importance of the interface management for light to heat conversion and provide guidelines on how to design future nanocomposites for the maximization (or minimization) of their photothermal response.

2. The fundamental mechanism behind the observed properties should be discussed. In Figure 2E, the positive correlation between QY and Q should be discussed. In figure 4, why do the Pd satellites interact weakly with light?

The positive correlation between QY and Q is related to the enhanced radiative recombination of charge carriers due to an increased density of photonic states (Purcell effect). It is not in the scope of this manuscript to provide a complete description of the mechanistic details for photoluminescence emission. Instead, the main message of Figure 2E is that the photoluminescence QY can also be used as an indicator of plasmon damping. In addition to references 66-70 and 72-75 of the revised manuscript, we have included the following two references for the readers that are particularly interested in deepening into the emission properties of bimetallic plasmonic NPs.

70. Zheng, Z., Tachikawa, T. & Majima, T. Single-particle study of Pt-modified Au nanorods for plasmon-enhanced hydrogen generation in visible to near-infrared region. *J. Am. Chem. Soc.* 136, 6870–6873 (2014).

71. Joplin, A. et al. Correlated Absorption and Scattering Spectroscopy of Individual Platinum-Decorated Gold Nanorods Reveals Strong Excitation Enhancement in the Nonplasmonic Metal. *ACS Nano* 11, 12346–12357 (2017).

3. The experimental process of AS has not been fully discussed. I wonder how the authors determine the relative position of the laser spot and the nanoparticle (Figure 3A)? If the laser keeps moving in a line, what is the spatial resolution of the spectroscopic measurements?

To improve the clarity, we have included the following sentence to the manuscript:

For each NP, a square raster scanning consisting in 10 x 10 pixels on a range of 0.8 x 0.8 μm^2 is performed. On each pixel, the PL spectra is measured with an integration time of 2.5 s. In such a manner, a set of one hundred temperature-dependent PL emission spectra is collected.

It must be noted that further description of the experimental process is provided in the section “Hyperspectral Confocal PL images” of the Materials and Methods and the section S6. Photothermal measurement of the supplementary materials. It should be stressed that the method is not an original contribution of this manuscript, and it was applied as fully described in Barella et. al. (*ACS Nano* 2021, 15, 2, 2458–2467). For this reason, we have decided to only include a short summary of the method in the main text.

4. Continuing 3, is there any shape deformation of nanoparticles during laser irradiance? This will lead to variations in the relative position of NPs and the laser.

No, the optical properties of the NPs remain unchanged during laser irradiance. In section S3 of the Supplementary Materials we present a stability study under different irradiances. The scattering spectrum of each NP before and after irradiation was compared. It was found that for the irradiances used for optical printing and photothermal characterization, the spectra remain unchanged, indicating that the NPs are stable and do not suffer from a light induced reshaping.

Minor issues:

1. Line 288: ‘weekly’ should be ‘weakly’

We thank the reviewer; it was changed accordingly.

2. In SM, line 283, “S11. Optical properties of Pd satellites”, should be erased. **The section merged into the new S11 of the revised Supplementary Materials.**

3. The authors used the seed-mediated method to form nanoparticles and CTAC as stabilizers. Usually, the shape of Au NPs via this method would be close to an octahedron, since CTAC prefers to attach to {111} facet. Would that affect the photothermal effect at different orientations? Did the authors apply NP aging to make smooth Au spheres?

We have not performed any aging or etching to the Au NPs. The synthetic route adopted to fabricate Au nanospheres (AuNS) was based on the method reported by Zheng et. al. (*Particle & Particle Systems Characterization* 31.2 (2014): 266-273) The only modifications introduced by us to the method were done pursuing the goal to enlarge the volume without affecting the dynamics of the system.

One of the most important aspects of this route is that the once that roughly 10 nm AuNS are synthesized, those can be later used as seeds to grow them until the desired size is reached (60 and 67 in our study). This growing step involves the pump of Au precursor solution (AuCl_4) at a low and controlled rate, which turns out crucial to preserve the sphericity of the resulting nanoparticles. As explained by Zheng et. al., there is an interplay between the reduction of precursor and the surface

diffusion of low-coordinated atoms. When the deposition of new atoms is low enough, the atoms with low coordination will diffuse across the surface to lower their energy, producing nanoparticles with spherical shape. If the rate of the deposition exceeds the diffusion rate, the spherical shape is highly compromised and the resulting particles are octahedron, as pointed out.

Therefore, when conducting our synthesis of AuNS, the deposition rate was adjusted to ensure sphere-like nanoparticles. This is confirmed by TEM images such as the ones presented in Figure 1 and Figure 4 of the revised manuscript.

The question regarding the role of facets in the photothermal response is indeed very interesting. In Section S10 of the revised Supplementary Materials, we address this point. Faceted NPs can present a larger contact area with the substrate, enhancing heat dissipation. It is shown that a faceted NPs with a planar contact with the substrate will present a photothermal coefficient 4% lower than a spherical NP touching the substrate in a single point.

Figure S14. A faceted Pd shell on a substrate simulated in COMSOL. The Pd shell is simulated as a (a) sphere and (b-c) an icosahedron. The icosahedron can be placed on the substrate with a single point of contact or a facet contact. (d) The temperature profiles from the center of the NPs in x direction.

Reviewer #6:

Studying plasmonic metals and plasmon-assisted catalysis is of far-reaching interest to the general scientific community. The authors reported the growth of colloidal Au/Pd nanoparticles with different core/shell structures and analyzed the impact of morphology and thickness on the photothermal properties of such hybrid structures. The authors have demonstrated that AS thermometry can provide information at a single particle level in a non-destructive way, which may help the understanding of other complex structure interfaces. Solid and insightful discussions are provided and I quite liked reading them. However, my main concerns reside in the sample quality and data analysis shown at the beginning of the manuscript.

We thank the reviewer for the overall positive feedback, and we provide below a detailed analysis of the sample and address all concerns.

The authors synthesized several Au/Pd core/shell structures, and used electron microscopy and/or EDS to identify their thickness and morphology. A very accurate thickness of the Pd shell is provided. For example, in line 127, the authors claim the Pd thickness are 2.4 ± 0.3 nm, and 3.6 ± 0.3 nm. My questions are how these numbers are calculated. The EDS shown in Fig. 1b looks very noisy, and the distribution of the Pd shell is obviously inhomogeneous. The authors should provide a detailed calculation process to repeat these results. In fig. 4a, the Pd shells all melt together and thickness is

again apparently inhomogeneous. It is quite surprising that a 0.3 nm error bar can be realized. In addition, many Au cores look not well faceted or irregular, but a 0.1 nm or 0.2 nm error bar is still given for the median size.

One interesting conclusion in this paper is that the photothermal coefficient is relatively irrelevant to the Pd shell thickness. Given the irregular Pd shell, I highly suggest the authors revisit the data and exclude the possibility of sample bias. In addition, the structural information of the Au/Pd interface is relatively unclear.

The thicknesses of the Pd shells were calculated from Transmission Electron Microscopy images, as follows:

The area A of each NP was measured, and a characteristic diameter was calculated as $d = 2\sqrt{\frac{A}{\pi}}$. At least 100 NPs were measured for each colloid, and the mean \bar{d} and standard deviation σ_d of the distributions were calculated. The standard deviations were in the range between 1 nm and 2 nm for all the colloids studied in the manuscript. The standard errors of the means were calculated as $\sigma_{\bar{d}} = \frac{\sigma_d}{\sqrt{N}}$ with N the sample size. These values were typically below 1 nm.

For Core Shells, the thickness t of the Pd shell was estimated as $t = \bar{d}_{CS} - \bar{d}_C$, where \bar{d}_{CS} and \bar{d}_C are the mean diameters of the Core Shell and Au Cores respectively. The error in the thickness t is the addition of the absolute standard errors of the mean diameters.

For the revised version of the manuscript, we have performed additional characterization of the Core Shell NPs using Inductive Coupled Plasma Atomic Emission Spectroscopy (ICP-AES). This technique allows the quantification of the ratio between the Au and Pd mass. Using this information, knowing the size of the Au cores, and assuming a perfectly spherical geometry for the cores and the CS, the thickness of the Pd shell can be estimated as

$$t = \sqrt[3]{\frac{3}{4\pi}(V_C + V_S)} - r_C = r_C \left(\sqrt[3]{1 + \frac{\rho_C M_S}{\rho_S M_C}} - 1 \right)$$

With $V_C = \frac{4}{3}\pi r_C^3$, $V_S = V_C \frac{\rho_C M_S}{\rho_S M_C}$. V is the volume, r is the radius, M is the mass and ρ is the density. Subscripts C and S refer to the core and the shell, respectively. The measured values are shown in the following table. For comparison, the results of the thickness characterization using the TEM method are included. We believe that the presented evidence excludes the possibility of sample bias.

	Core Size (TEM)	Pd/Au $\frac{M_S}{M_C}$	Pd Thickness (ICP-AES)	Pd Thickness (TEM)
Au67@Pd2	(66.6 ± 0.2) nm	(0.105 ± 0.005)	(1.8 ± 0.4) nm	(2.4 ± 0.3) nm
Au67@Pd4	(66.6 ± 0.2) nm	(0.21 ± 0.01)	(3.4 ± 0.4) nm	(3.6 ± 0.3) nm
Au60@Pd2	(59.7 ± 0.2) nm	(0.073 ± 0.004)	(1.1 ± 0.4) nm	(1.8 ± 0.3) nm

We thank the reviewer for pointing towards Figure 4a. The light grey material that appears to “melt together” is arguably the CTAC surfactant and not the Pd. We recognise that the Figure was indeed

misleading, so we performed additional TEM imaging of the CS NPs after removing the surfactant with ethanol.

The revised version of the Figure is shown here.

Figure 4: Optical and photothermal comparison between core-shell and core-satellites AuPd nanoparticles. **A)** Illustration and TEM images of the synthesized NPs. Scale bars: 20 nm. **B)** Experimental extinction spectra. Each spectrum is normalized using its own maximum. **C)** Average of single particle PL emission spectra of NPs on glass substrates and immersed in water, excited with 1 mW/ μm^2 of laser light at 532 nm. Grey band with no data corresponds to the laser rejection filter. **D)** Histograms of the experimental measured photothermal coefficient β for the three systems on glass substrates and immersed in water, at 532 nm.

A new section S1 was added to the revised version of the Supplementary Materials that contains a summary of the size characterizations and a detailed description of the calculation processes. Also, the following sentence was added to the manuscript to reflect the previous discussion:

“In addition, Inductive Coupled Plasma Atomic Emission Spectroscopy (ICP-AES) was employed to quantify the Pd to Au mass ratio of each CS-NPs colloid. Using this information, the thicknesses of the Pd shells were estimated to be (1.8 ± 0.4) nm, and (3.4 ± 0.4) nm, as described in the section S1 of the supplementary materials (SM).”

Pd is grown epitaxially on Au or what is the crystallographic information along the interface region.

We thank the reviewer for this question. Core-shell nanoparticles were produced by reducing the Pd precursor onto the Au cores, which served as template, using ascorbic acid (reducing agent) in the presence of CTAB. In prior studies in which the same procedure was used to achieve similar structures, it was found that the Pd shell presents a crystalline phase, whose growth seems to be unaffected by the core. (Fan, Feng-Ru, et al. "Epitaxial growth of heterogeneous metal nanocrystals:

from gold nano-octahedra to palladium and silver nanocubes." *Journal of the American Chemical Society* 130.22 (2008): 6949-6951.).

In order to corroborate that a similar behaviour is occurring in our system, we employed Scanning Transmission Electron Microscopy (STEM) in High Angle Angular Darkfield (HAADF) mode and in Bright Field mode. We have included the following text and figure to the section S1 of the supplementary materials.

Crystallinity of the Pd shell was studied Scanning Transmission Electron Microscopy (STEM) in High Angle Annular Darkfield (HAADF) mode and in Bright Field mode. The Figure S2 shows two images corresponding to the Au₆₇@Pd₄ system. The images indicate that growth of Pd on the Au core is crystalline.

Figure S2. Crystallinity of Core-shell nanoparticles. Scanning Transmission Electron Microscopy in High Angle Annular Darkfield (a) and Bright Field (b) mode. Images correspond to a Au₆₇@Pd₄ CS-NP.

Overall, this is a nice paper that contains many high-quality experimental results. The discussion is relatively clear. However, careful calculations and insightful discussions are suggested to allow the manuscript to be published in a high-profile journal like *Nat. Commun.*

There are some minor grammatical errors and although none interfere with understanding and reading. Careful proofreading is recommended. Here are some minor typos or errors listed below:

Line 255, change constant to constants
Line 369, remove additional space before the comma
Line 372, remove additional space before the comma, add in one space before ammonium hydroxide.

We thank the reviewer for finding these typos, they were corrected accordingly.

Reviewer comments, second round

Reviewer #1 (Remarks to the Author):

In this revised version, regarding my concern on the estimation of the Pd content on Au NP, there are more inconsistencies than in the previous version.

1- The authors considered a Pd-sat size dispersion of 6.3 ± 0.2 nm. However, from the ICP-AES measurements, they measured 6.3 ± 1.3 . Why are they here considering a dispersion of only 0.2 nm, and not 1.3, as measured? Then, the Pd-sats on the size image are far from being as big as 6 nm (the biggest is around 4 nm) and far from being monodispersed (1 to 4 nm I would say). So the size distribution in the sample they measured by ICP-AES does not look like the size distribution that ends up on the NPs. It looks obvious from the SEM images, but the authors don't say anything on this subject (see the image I attached to my review).

2- this alpha of 34° is overestimated. The authors are saying that on the SEM image of Figure 4A, they are seeing 54% of the Pd-sats covering the AuNP. It does not look the case (see the image I attached to my review). The Pd-sats being at 33° , 32° , 31° , 30° won't obviously be seen either. I would have said that at least half of the satellite has to be visible to be identified, which would strongly reduce the alpha value, and would multiply the estimated content of Pd-sats by a factor of ~ 2 . But even this consideration does not look appropriate because, on the SEM images, I cannot identify any nanoparticle that is partially masked by the Au core. This arbitrariness makes the error bar on this estimation huge. I would suggest the authors to better estimate (more appropriate Pd-sat diameter, better alpha angle), or just forget this way of estimating the Pd content, and only stick to ICP-AES estimation of the Pd/Au ratio, which is all what matters at the end.

3- In Figure 4D, the fact that the plasmon resonance is not shifted with AuNP-Pd-sat may just be that the amount of Pd is much less compares with CS AuPd NP. Line 297, the authors should also notice that.

Reviewer #2 (Remarks to the Author):

I have carefully revised the response letter of the authors, as well as all the new pieces of information added to the manuscript. I must congratulate them on their very thorough work. Undoubtedly they took all the comments seriously and worked to improve the manuscript. On my side, I think the answer is quite complete. The characterization of the particles is really good now, considering number of satellites and their distance to the particle. Regarding the diverse results found in the literature finding blue or red shift upon the addition of Pd, if not totally explained (which is understandable given that the other papers may not provide all the information needed for that), the information provided now is clearly enough to assure a complete description. This same philosophy of trying to provide information that helps building general theories and allow for future comparison, made me ask for some way of quantifying the thermal increase that can be extrapolated. The thermal coefficient here doesn't allow for that, as it depends on the thermal dissipation properties of the environment. Instead, photothermal efficiency (or absorption efficiency, whichever) does not, as it is intrinsically normalized. So, in principle, I was thinking on applying some sort of normalization here as well. Still, I understand that the experimental scheme makes it difficult and thus, calculating heating efficiency is probably a reasonable way of reaching a compromise.

Reviewer #3 (Remarks to the Author):

I have read the authors' reply to my and the other reviewers' comments. My minor comments

have been satisfactorily answered and I found the replies to the main observations of the other reviewers convincing. Therefore, I reiterate my appraisal of the scientific validity and high quality of this manuscript, and I support the editors' decision if they offer publication.

Reviewer #4 (Remarks to the Author):

The authors have adequately addressed the concerns raised in my first review. I recommend publication.

Reviewer #5 (Remarks to the Author):

The revised version is good with all my questions well addressed.

One minor issue is that the authors have answered my questions on 'if their results are generally applicable for other bimetallic systems', but they did not add the relevant contents in the manuscript 'Discussion part'. I would suggest the authors doing so.

The paper has reached the quality of NC and can be published after this minor revision.

Reviewer #6 (Remarks to the Author):

The authors have clearly addressed all the concerns from referees, and I think this work is ready to be accepted.

Response to reviewers:

Here we provide a point-by-point response to each comment, **in bold font**, indicating when opportune the modifications made to the manuscript (in blue).

Reviewer #1:

In this revised version, regarding my concern on the estimation of the Pd content on Au NP, there are more inconsistencies than in the previous version.

1- The authors considered a Pd-sat size dispersion of 6.3 ± 0.2 nm. However, from the ICP-AES measurements, they measured 6.3 ± 1.3 . Why are they here considering a dispersion of only 0.2 nm, and not 1.3, as measured? Then, the Pd-sats on the size image are far from being as big as 6 nm (the biggest is around 4 nm) and far from being monodispersed (1 to 4 nm I would say). So the size distribution in the sample they measured by ICP-AES does not look like the size distribution that ends up on the NPs. It looks obvious from the SEM images, but the authors don't say anything on this subject (see the image I attached to my review).

The authors claim that the Au NP is covered with a monodispersed layer of Pd NP (6 ± 0.2 nm in diameter). The largest Pd particle I could see on this image is only 4 nm big and the dispersion in size looks much larger (1 to 4 nm I would say).

Here is what the authors see, 27 Pd NP covering a region close to the equator of the Au NP, all of them 6 nm in diameter, to be compared with image A. It does not look to properly render reality.

We would like to thank reviewer 1 for the constructive comments and the detailed inspection of our data. The size characterization of the Pd satellites was performed using TEM microscopy. Motivated by the reviewer's question, we have increased the number of measured NPs. The revised version of the supplementary materials has a new Figure 15, which is the following.

Figure S15. Size characterization of Pd satellites. (a) Histogram of the diameter of Pd Satellites. The median diameter is $d = (6.2 \pm 0.3) \text{ nm}$ with a standard deviation of $\sigma_d = 1.3 \text{ nm}$. (b) Histogram of the volume of Pd Satellites. The median volume of the Pd satellites was estimated to be $V = (125 \pm 7) \text{ nm}^3$ with a standard deviation of $\sigma_V = 120 \text{ nm}^3$. (c) TEM images of the Pd satellites.

The size distribution is shown in Figure S15a. The satellites are, as the reviewer notes, not monodisperse, with diameters ranging from 4 to 8 nm. The standard deviation is $\sigma = 1.3 \text{ nm}$.

In the revised main manuscript, we inform the measured size of the Pd satellites in this way:

The satellites have a median diameter of $(6.2 \pm 0.3) \text{ nm}$ and a standard deviation of $\sigma = 1.3 \text{ nm}$

It must be noted that the error of 0.3 nm is the standard error of the mean, calculated as $\sigma_{\bar{d}} = \frac{\sigma_d}{\sqrt{N}}$ with N the sample size. The standard error of the median is typically smaller, so we used the standard error of the mean as an upper bound.

The size distribution of the revised Figure S15a was obtained from the TEM images shown in Figure S15c. The imaging conditions were optimized to properly see the Pd satellites, enhancing the contrast of Pd with the maximum possible magnification. The reviewer is right when noting that the some of the Pd satellites look smaller on the TEM images of Figure S16. This could be due to the

following reasons: i) The imaging conditions were optimized to see simultaneously Au and Pd. The main goal of these images was to show that the satellites were correctly assembled on the Au core, and therefore have a lower magnification and a worse contrast of Pd than the ones from Figure S15c. ii) Many Pd satellites are partially masked by the Au cores, resulting in a smaller visible size.

For these reasons, we believe that the correct size determination of the satellites should be done with the TEM images of the Pd satellites before their assembly to the Au cores, and that is what we have done.

2- this alpha of 34° is overestimated. The authors are saying that on the SEM image of Figure 4A, they are seeing 54% of the Pd-sats covering the AuNP. It does not look the case (see the image I attached to my review). The Pd-sats being at $33, 32, 31, 30^\circ$ won't obviously be seen either. I would have said that at least half of the satellite has to be visible to be identified, which would strongly reduce the alpha value, and would multiply the estimated content of Pd-sats by a factor of ~ 2 . But even this consideration does not look appropriate because, on the SEM images, I cannot identify any nanoparticle that is partially masked by the Au core. This arbitrariness makes the error bar on this estimation huge. I would suggest the authors to better estimate (more appropriate Pd-sat diameter, better alpha angle), or just forget this way of estimating the Pd content, and only stick to ICP-AES estimation of the Pd/Au ratio, which is all what matters at the end.

We thank the reviewer for making this point. We have decided to follow the reviewer's suggestion and remove the estimation of number of satellites based on counting satellites in the TEM images. Because the current version has a more accurate way of quantifying the number of satellites based on ICP-AES, the counting method is no longer needed. In the revised version of the SM, only the method based on ICP-AES is described. This change is reflected in the revised main manuscript as:

The average number of satellites per NP of the Au60-Pd-sat was estimated to be $\langle N_s \rangle = (54 \pm 7)$.

The new informed value is the one obtained by the ICP-AES, and the "counting method" is no longer considered in the calculation.

However, we would like to comment that the inconsistencies between both methods are not as large as the reviewer suggested. According to the ICP-AES method, the number of satellites is $N_s = (54 \pm 7)$. Using the counting method with a value of $\alpha_T \approx 34^\circ$, we obtain $N_s = 48$. If we follow the reviewer suggestion and consider that at least half of the satellite must be visible to be identified, we obtain $\alpha_T \approx 20^\circ$ and $N_s = 77$. While it is true that $\alpha_T \approx 34^\circ$ might be overestimated, $\alpha_T \approx 20^\circ$ might be underestimated, and the number of satellites lays in the range $48 < N_s < 77$, not that far from the more accurate measurement of $N_s = (54 \pm 7)$ given by the ICP-AES method.

3- In Figure 4D, the fact that the plasmon resonance is not shifted with AuNP-Pd-sat may just be that the amount of Pd is much less compares with CS AuPd NP. Line 297, the authors should also notice that.

The number of satellites does not have a strong influence on the position of the resonance for the Au60-Pd-sat system. It has been shown that the calculated position of the maximum remains unchanged when increasing the Pd load from 50 to 150 satellites. (Herran et al., Tailoring Plasmonic Bimetallic Nanocatalysts Toward Sunlight-Driven H₂ Production. *Adv. Funct. Mater.* 2022, 32, 2203418. <https://doi.org/10.1002/adfm.202203418>). The small 3 nm redshift and slight broadening of the Au60-Pd-sat spectrum with respect to the Au NS 60 that is shown in Figure 4B is consistent with previous calculations by Herran *et al.* In addition, Figure 4C of the main manuscript shows a significantly lower photoluminescence emission of the Au60-Pd-sat in comparison with the Au NS 60, caused by the presence of Pd satellites. Finally, it must be noted that the total Pd mass in the Au60-Pd-sat system is similar to the mass of the Au60@Pd2 CS, as confirmed by ICP-AES.

Reviewer #2:

I have carefully revised the response letter of the authors, as well as all the new pieces of information added to the manuscript. I must congratulate them on their very thorough work. Undoubtedly they took all the comments seriously and worked to improve the manuscript.

On my side, I think the answer is quite complete. The characterization of the particles is really good now, considering number of stallites and their distance to the particle. Regarding the diverse results found in the literature finding blue or red shift upon the addition of Pd, if not totally explained (which is understandable given that the other papers may not provide all the information needed for that), the information provided now is clearly enough to assure a complete description.

This same philosophy of trying to provide information that helps building general theories and allow for future comparison, made me ask for some way of quantifying the thermal increase that can be extrapolated. The thermal coefficient here doesn't allow for that, as it depends on the thermal dissipation properties of the environment.

Instead, photothermal efficiency (or absorption efficiency, whichever) does not, as it is intrinsically normalized. So, in principle, I was thinking on applying some sort of normalizarion here as well. Still, I understand that the experimental scheme makes it difficult and thus, calculating heating efficiency is probably a reasonable way of reaching a compromise.

We thank reviewer 2 for the careful revision and the insightful discussions, which made us think and improve the manuscript. We agree with the summary of the discussion presented by the reviewer. The photothermal coefficient β (i.e. the relation between light irradiance and temperature increase) does depend on the thermal dissipation properties of the environment. However, we would like to note that the role of the glass substrate in heat dissipation is minor (see sections S9 and S10 of the SI, and our previous work Barella *et al.* ACS Nano 2021, 15, 2, 2458–2467). For this reason, the informed photothermal coefficients β are useful to estimate the photothermal response of these individual NPs in water solutions.

Reviewer #3:

I have read the authors' reply to my and the other reviewers' comments. My minor comments have been satisfactorily answered and I found the replies to the main observations of the other reviewers convincing. Therefore, I reiterate my appraisal of the scientific validity and high quality of this manuscript, and I support the editors' decision if they offer publication.

We thank reviewer 3 for the appraisal of the work and the acceptance of the manuscript.

Reviewer #4:

The authors have adequately addressed the concerns raised in my first review. I recommend publication.

We thank reviewer 4 for the acceptance of the manuscript.

Reviewer #5:

The revised version is good with all my questions well addressed.

One minor issue is that the authors have answered my questions on 'if their results are generally applicable for other bimetallic systems', but they did not add the relevant contents in the manuscript 'Discussion part'. I would suggest the authors doing so.

We thank the reviewer for the suggestion. We have modified a paragraph in the discussion to address this point. The revised version of the manuscript has the following paragraph:

Since Pd is highly conductive, it does not significantly restrict the conduction of heat to the surroundings, meaning that the absorption cross section of the Au-Pd CS-NPs is the main parameter

influencing its photothermal coefficient. This conclusion can be extended to other CS-NPs where both materials are highly conductive. (See section S14 of the SM for a discussion on Au@Pt, Au@Rh and Au@Ag NPs).

The paper has reached the quality of NC and can be published after this minor revision.

We thank reviewer 5 for the acceptance of the manuscript.

Reviewer #6:

The authors have clearly addressed all the concerns from referees, and I think this work is ready to be accepted.

We thank reviewer 6 for the acceptance of the manuscript.